# Electro-capillary peeling of thin films

**Peiliu Li** [1,2], **Xianfu Huang** [1,2] **& Ya-Pu Zhao** [1,2] ✉

Thin films are widely-used functional materials that have attracted much interest in academic and industrial applications. With thin films becoming micro/nanoscale, developing a simple and nondestructive peeling method for transferring and reusing the films remains a major challenge. Here, we develop an electro-capillary peeling strategy that achieves thin film detachment by driving liquid to percolate and spread into the bonding layer under electric fields, immensely reducing the deformation and strain of the film compared with traditional methods (reaching 86%). Our approach is evaluated via various applied voltages and films, showing active control characterizations and being appropriate for a broad range of films. Theoretically, electro-capillary peeling is achieved by utilizing the Maxwell stress to compete with the film's adhesion stress and tension stress. This work shows the great potential of the electro-capillary peeling method to provide a simple way to transfer films and facilitates valid avenues for reusing soft materials.

Thin films are universal functional base materials with a variety of outstanding properties that have been widely investigated and used in fundamental studies and applications; these applications include flexible electronics[1,2], soft robotics[3,4], micro/nanoelectromechanical systems (MEMS/NEMS)[5,6], and biomedical devices[7,8]. In general, thin films serve as base materials implanted by some functional structures and devices to achieve a specific ability. For example, carbon nanotube thin-film transistors are promising candidates for flexible and wearable electronics[9,10], thin films with a crystalline can be programmed into various soft robotics[11,12], and thin films can be planted with ZnO nanorods to fabricate a self-cleaning surface[13,14]. Because thin films and implanted structures/devices often have greatly different mechanical properties, these multilayered thin film systems are often prone to interfacial delamination/cracking under various loading conditions, resulting in premature failure of the devices. Therefore, investigating a nondestructive peeling method for thin films is highly attractive for the long-term reliability of functional devices in applications.

To achieve the miniaturization of the components, the thickness of the thin film used has increasingly become smaller and is on the micro/nanoscale level. More recently, the capability of micro/nanofilms has presented exciting opportunities in materials design, such as assembling multiple two-dimensional (2D) materials with complementary properties into layered heterogeneous structures[15,16], superconducting films for electronic devices[17], and supercapacitor electrodes in energy storage[18,19]. However, traditional peeling methods, including the debonded strip[20] and blister method[21], have difficulty satisfying the needs of practical production at these scales, especially for fully attached micro/nanofilms. Based on practical applications, several strategies have been pursued to manipulate the conditions for detaching the micro/nanofilm from the substrate, including hydrogen bubbles evolved at alkaline water electrolysis for graphene on the metallic catalyst[22], drying-induced peeling for colloidal films[23], water-soluble chemical bonding layer[24], and the reduction of surface adhesion modified by chemical/physical approaches[25–28]. Generally, these peeling strategies exploit modifying the bonding layer's properties between the film and substrate. Despite extensive progress, the development of peeling methods for micro/nanofilms remains in its infancy, owing to the complicated preparations and the limitations of applied films. Therefore, there is an urgent need to develop a simple peeling method for various thin films.

In contrast to complicated modifications of the bonding layer's properties, a thin liquid layer percolating and spreading into the bonding layer can serve as a simple physical peeling method for detaching the film from the substrate. The capillary peeling approach is a passive method that utilizes the liquid layer to detach an attached film when the film-substrate interface contacts water[29,30]. The principle of the capillary peeling method illustrates that it is generally suitable for films with low surface adhesion, especially hydrophobic films. In practice, there are many films with a high adhesive force, and the peeling rate in an on-demand manner is highly preferred; however, this

---

[1]State Key Laboratory of Nonlinear Mechanics, Institute of Mechanics, Chinese Academy of Sciences, Beijing, China. [2]School of Engineering Science, University of Chinese Academy of Sciences, Beijing, China. ✉e-mail: yzhao@imech.ac.cn

necessitates stricter requirements on the manipulation design. To this end, developing an active peeling method that enables easy detachment of various micro/nanofilms remains a major challenge. Previous studies[31–33] have shown that the wettability of a liquid can be controlled by electric fields, including surface tension, contact angle, and movement of the contact line. The electric charge in the liquid is loaded by the Maxwell stress, which impacts the motion of the liquid[34–36]. Both the wettability and motion of the liquid are controlled by the electric field. Therefore, developing a peeling method by controlling a thin liquid layer to detach micro/nanofilms under an electric field is potentially a valid option.

Herein, we develop an electro-capillary peeling method for thin film detachment that is achieved by driving liquid to percolate and spread into the bonding layer under an electric field. Unlike the complicated modification of the bonding layer and applied film restriction of capillary peeling, the electro-capillary peeling method is an active detaching approach with a direct current (DC) supply power and can be applied to a broad range of films, even in hydrogels of high adhesion stress. Theoretically, the electro-capillary peeling method is achieved by the Maxwell stress competing with the film's adhesion stress and tension stress. The peeling length versus time, applied voltage, film thickness, and elastic modulus are described systematically by $r \sim t$, $r \sim U$, $r \sim d_0^{-2}$, and $r \sim E^{-2}$, respectively. In addition, observations of the critical peeling voltage indicate that a polydimethylsiloxane (PDMS) film (thickness of 100 μm and elastic modulus of 1.0 MPa) is easily detached from indium tin oxide (ITO) glass by using the electro-capillary peeling method at a voltage of 0.7 V, and the thin film deformation indicates that the electro-capillary peeling method is a nondestructive approach for functional device protection. Therefore, the electro-capillary peeling method has promising applications in detaching and transferring thin films.

## Results

### Experimental apparatus and characterization of the electro-capillary peeling

Figure 1a shows a schematic drawing of the electro-capillary peeling method that mainly includes three core elements: a film, an electrolyte droplet, and a supply power. Initially, the PDMS film served as tested samples due to its stable property and extensive application, and it of a micron thickness was fabricated by using the spin coating method (see Methods). After being treated by the plasma, the PDMS films were bonded on the ITO glass surface with a firm and uniform bonding layer (Supplementary Fig. 1). The wettability showed that the bonded surface was hydrophilic, and the water blister test[37] (see Methods) presented that the wet adhesion of the bonding layer was approximately 133 mJ m⁻² (Supplementary Fig. 2 and Supplementary Table 1). Thus, there was a strong combination between the PDMS films and ITO glass surfaces. Furthermore, a circular hole with a radius of 2 mm was cropped out at the center of the bonded PDMS film. A micropump injected an electrolyte droplet with a volume of 20 μL into this circular hole, and then an electric field was applied to the liquid droplet by a DC supply power with two platinum (Pt) wire electrodes. One of the Pt wire electrodes was inserted into the liquid droplet and connected to the positive pole, and the other Pt wire electrode was fixed on the ITO glass surface and attached to the negative pole. After the power was turned on, the liquid droplet percolated into the bonding layer and detached the PDMS film from the ITO glass (Fig. 1b and Supplementary Movie 1). Viewed from side views, the peel-off film's profile was similar to a parabolic shape (Fig. 1c). As observed from top views, the wetting shape was an axisymmetric circle, and the ring's radius $r(t)$ was the peeling length, which increased with time (Fig. 1d). Under electric fields, the liquid was driven to percolate and spread into the bonding layer for film detachment.

### Characterizations of the electro-capillary peeling at different voltages

To explore the impact of electric intensity on the electro-capillary peeling method, the lifting height and peeling length were observed over broad ranges of applied voltages. As shown in Fig. 2a, side views of the electro-capillary peeling showed that the lifting height of the PDMS film was similar for the voltage range from 1.5 to 4.5 V. Yet, the droplet outline was quite different at the various voltages. With an increasing applied voltage, the topography of the droplet changed from a semicircle shape to a half-ellipse shape due to the increased peeling length (Supplementary Movie 2). Figure 2b, c show top views of electro-capillary peeling in axisymmetric and planar peeling modes. These two

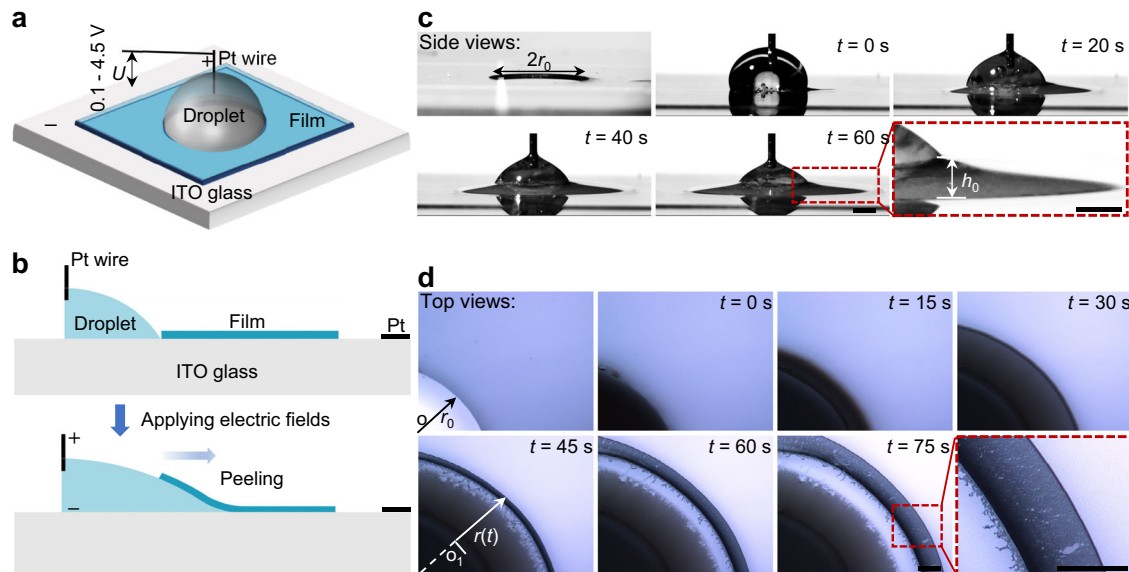

**Fig. 1 | Experimental setup and characterization of electro-capillary peeling.**
**a** Experimental apparatus setup. The electric field is applied through two Pt wire electrodes; one of the Pt wire electrodes is inserted in the liquid droplet and the other is fixed on the ITO glass. **b** Schematic illustration of the working process in the electro-capillary peeling method. The liquid droplet is driven to percolate into the bonding layer and peel off the PDMS film from the ITO glass by applying a DC electric field. **c**, **d** Side and top views of the electro-capillary peeling process. $r_0$ is the radius of the prefabricated hole on the PDMS film, $h_0$ is the lifting height at the edge of the film, and $r(t)$ is the time-dependent peeling length. Here, all scale bars are 1 mm.

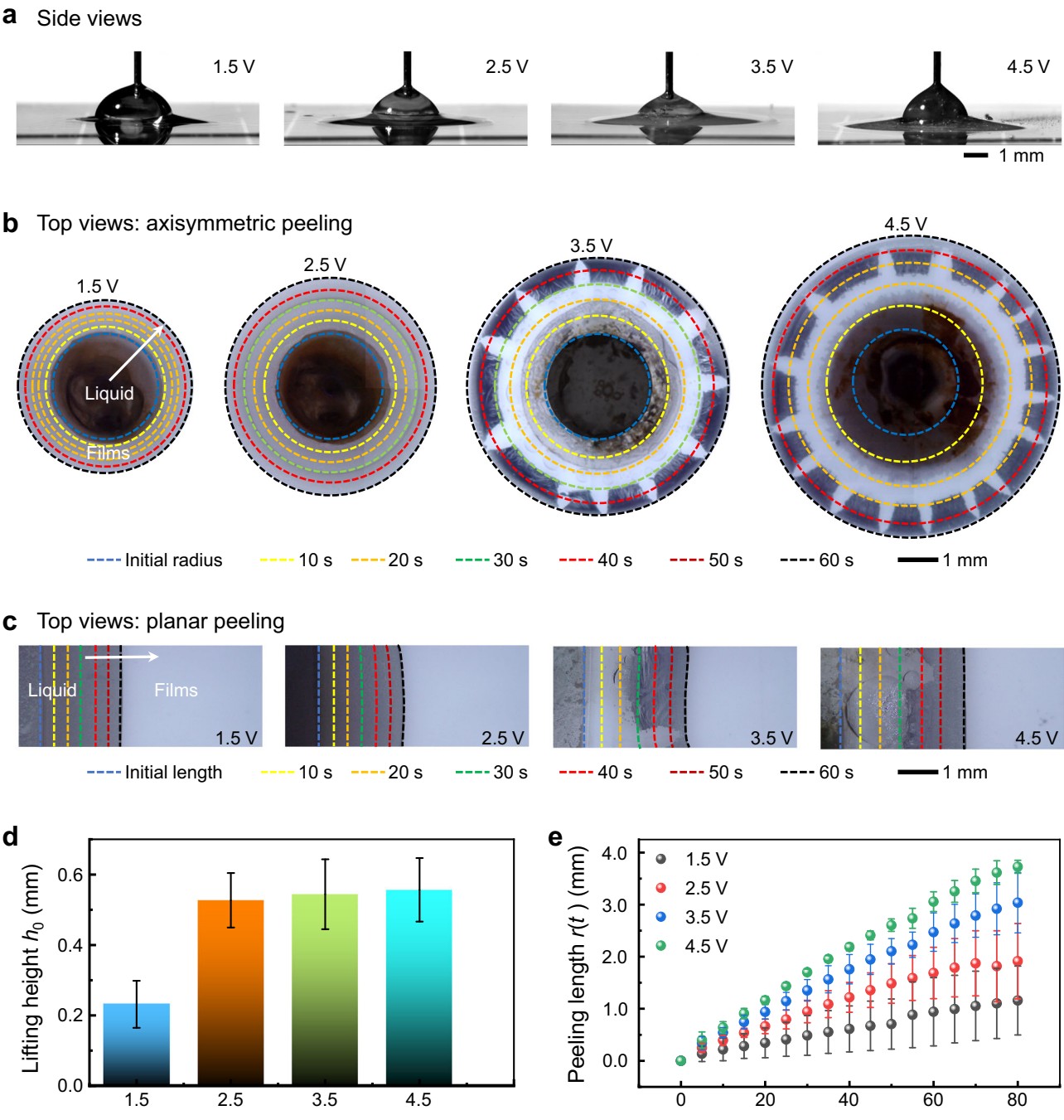

**Fig. 2 | Electro-capillary peeling at different voltages. a** Side views of electro-capillary peeling at various voltages. **b, c** Top views of axisymmetric and planar peeling at various voltages. The selected photo shows the end of electro-capillary peeling ($t = 60$ s), and the dotted lines with different colors are used to mark its evolution. The "fingering" graphics are the pattern formed by the potassium chloride (KCl) precipitation. **d, e** Statistical results of the lifting height and peeling length in the electro-capillary peeling method. At 1.5, 2.5, 3.5, and 4.5 V, the lifting height is 0.23, 0.51, 0.54, and 0.56 mm, and the peeling length is 1.1, 1.9, 3.0, and 3.8 mm ($t = 80$ s), respectively. The peeling length data are collected within 80 s because a liquid droplet of 20 μL would completely spread in some bonding layers at this time. Error bars in **d, e** are the standard deviation of the raw data.

modes are the main types in practical applications[38], and their experimental setups are shown in Supplementary Fig. 3. At 1.5 V, the peeling length was very small and even smaller than the radius of the pre-fabricated hole ($t = 60$ s); as the applied voltage increased to 4.5 V, the peeling length was almost three times that at 1.5 V. In these two modes, observation of peeling length showed that both their efficiency and manner are similar (Supplementary Movie 3), and the peeling rate increased with increasing applied voltage. Statistical results of the lifting height and the peeling length at different voltages are presented in Fig. 2d, e. When the applied voltage changed from 1.5 to 2.5 V, the lifting height changed by 122% (from 0.23 to 0.51 mm). As the applied voltage increased above 2.5 V, the lifting height slightly varied with increasing voltage (Fig. 2d). Figure 2e shows the peeling length versus time at the various voltages. At the same peeling time, the peeling length evidently increased with increasing applied voltage. The peeling length-time curve further showed that the peeling length was nearly linear with time.

At voltages of 1.5, 2.5, 3.5, and 4.5 V, the average peeling rates were 0.013, 0.023, 0.038, and 0.048 mm s$^{-1}$, respectively. The peeling rate showed a linear relationship with the applied voltage. Therefore, depending on the practical application, the efficiency of electro-capillary peeling could be actively controlled by the applied electric field.

**Electro-capillary peeling method for various films' detachment**
In practice, there are many films with various properties used in different applications. To evaluate the practicality of the electro-capillary peeling method, systematic experiments were conducted with films of different thicknesses, elastic moduli, and types. Initially, PDMS films with thicknesses from 25 to 300 μm were fabricated by a designed spin coating rate (see Methods). A representative SEM image of the PDMS film on the glass is shown in Fig. 3a, and the PDMS films with thicknesses from 25 to 300 μm are shown in Supplementary Fig. 4. The PDMS films fabricated using the spin coating method had a uniform thickness and well bonded on the glass. Furthermore, the PDMS films with the elastic moduli of 1.0, 1.6, and 2.4 MPa were achieved by consisting of the intentional weight ratio of the base and curing agent (see Methods). Their force-displacement curves indicated that the film was at the elastic stage within the tensile range of 25 mm (Fig. 3b). The water blister test shown in Supplementary Fig. 2 and Supplementary Tables 1 and 2 showed that the wet adhesion of these films on the ITO glass was almost the same regardless of the thin film thickness and elastic modulus. Ultimately, the functional films, including hydrogel, polyethylene terephthalate (PET), and polyethylene naphthalate (PEN) film, were also tested by using the electro-capillary peeling method. SEM images showed that the bonding layer of these functional materials was the same as that of the PDMS films (Fig. 3c). Statistical results of the lifting height and peeling length for various films are provided in Fig. 3d, e. Both the lifting height and peeling length decreased with increasing the film thickness. When the film thickness changed from 25 to 300 μm, the lifting height and peeling length were reduced from 0.62 to 0.13 mm and from 3.61 to 0.97 mm, respectively. The observations of the electro-capillary peeling with the films of various elastic moduli were similar to those of thickness. The lifting height and

peeling length decreased 89.3% and 64.0%, respectively, when the elastic modulus increased from 1.0 to 2.4 MPa. For a thicker or harder film, the electro-capillary peeling rate would be slower and potentially reduced to be disregarded (Supplementary Fig. 5). For widely used functional films (hydrogel, PET, and PEN film), our electro-capillary peeling method was able to detach them from the substrate, although they had different elastic moduli and wet adhesion (Supplementary Tables 3, 4). The lifting height of the hydrogel was the largest, and that of PET was the smallest; the peeling length of the hydrogel was the smallest, and that of PEN was the largest. Although the performance of the electro-capillary peeling method was affected by the film's properties, this approach was clearly appropriate for a broad range of films.

**Theoretical model and dynamic analysis of the electro-capillary peeling**
To clearly understand the electro-capillary peeling observed in the experiment and to further explain the impact of the applied voltage and film properties on the detachment behavior, we establish a theoretical model from the perspective of force analysis. In this work, a film with a thickness of $d_0$ is bonded on the ITO glass surface. At the pre-fabricated hole on the film, a liquid droplet with a density of $\rho$, surface tension of $\gamma$, relative permittivity of $\varepsilon_f$, and ion density of $\rho_f$ is placed (Fig. 4a). Because the working characteristic length of the liquid layer thickness ($10^{-4}$ m) in this work is much smaller than that of the Bond number ($10^{-3}$ m)[34], the effect of fluid gravity can be neglected (corresponding to Fig. 2d). For the axisymmetric electro-capillary peeling mode, a unit model with a width of $l_0$ is analyzed in polar coordinates. $r(t)$ and $w(r)$ are the peeling length and the deflection of the film, respectively. Calculations are based on the theory of elasticity[39], and the deflection of the film is described as

$$T \frac{\mathrm{d}^2 w}{\mathrm{d}x^2} = p \tag{1}$$

where $p$ is the net pressure loading on the film and $T$ is the film tension stress. At a strain of $\varepsilon$, $T = E\varepsilon d_0$, where $E$ is the elastic modulus

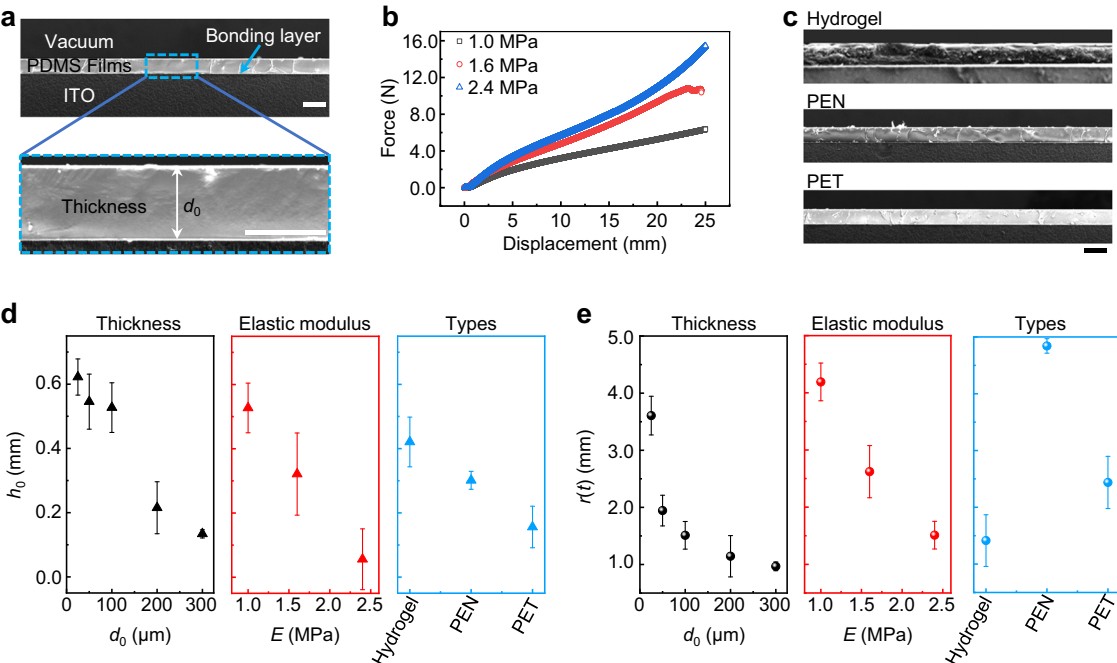

**Fig. 3 | Electro-capillary peeling for various films. a** Representative SEM image of the PDMS film on the glass surface. $d_0$ is the film's thickness. **b** Force–displacement curves of the film with different elastic moduli. The thickness and width of the tested film are 1 mm and 20 mm, respectively. **c** SEM images of the functional films

on the glass surface. The thicknesses of the hydrogel, PEN, and PET films are all 100 μm. **d, e** Statistical results of the lifting height and peeling length for the films with diverse thicknesses, elastic moduli, and types. Error bars in **d** and **e** are the standard deviation of raw data, and all scale bars in **a** and **c** are 100 μm.

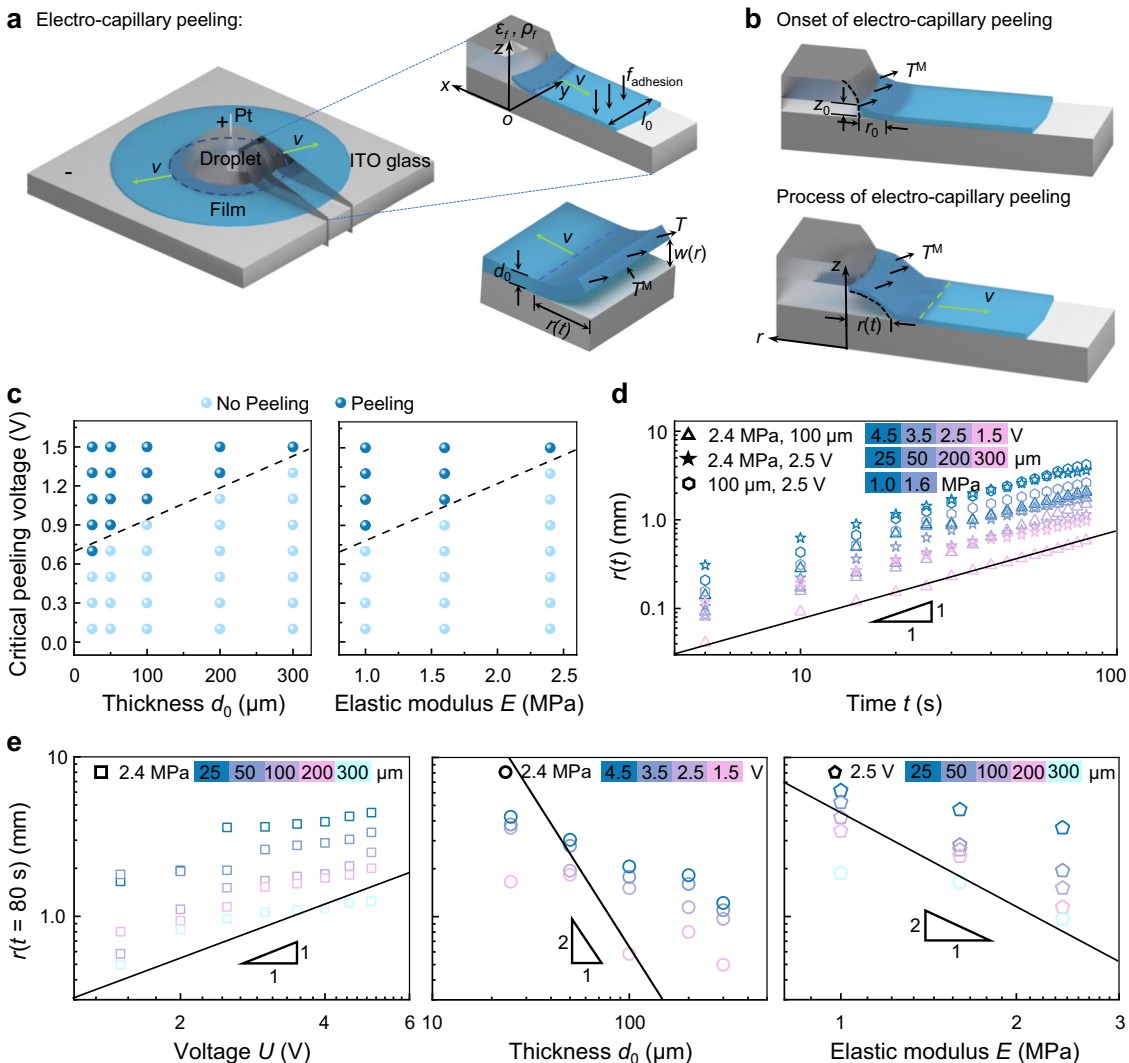

**Fig. 4 | Working mechanism of electro-capillary peeling. a** Schematic illustration of the electro-capillary peeling mechanism. A unit model with a width of $l_0$ is analyzed in orthogonal coordinates. $x$, $y$, and $z$ are the radial, tangential, and film thickness directions, respectively. **b** Electro-capillary peeling process. At the onset of electro-capillary peeling, the liquid droplet detaches the film with an ultralow lifting height of $z_0$ and does not wet the bonding layer. As the peeling proceeds, the lifting height increases, and the liquid droplet wet and spreads into the bonding layer. The droplet profile is marked by a dotted line. **c** Critical peeling voltage versus thin film thickness and elastic modulus. The critical peeling voltage versus thickness and elastic modulus are tested with PDMS film of 1.0 MPa and 100 μm, respectively. Dark and light blue spheres show that films can be peeled and not be peeled at such voltages. **d** Peeling law of the electro-capillary peeling versus time. Solid line: $r(t) \sim t$. **e** The impact of the applied voltage and film properties on the peeling length. Solid line: $r(t) \sim U$, $r(t) \sim d_0^{-2}$, and $r(t) \sim E^{-2}$. All experiments were repeated at least 6 times.

of the film. Under various electric fields, the net pressure loading on the film is obtained from the equilibrium relationship between the film's dead weight and the electric force, which is expressed as $p = T^M - \rho_m g d_0$, where $T^M$, $g$, and $\rho_m$ are the Maxwell stress, gravitational acceleration, and film density, respectively. According to the Korteweg-Helmholtz law[40], the Maxwell stress is expressed as $T^M = \varepsilon_f \rho_f E_e$, where $E_e$ is the electric intensity. Due to the film with a micron thickness, the pressure induced by the film's dead weight can be neglected compared with the Maxwell stress. Substituting the net pressure into Eq. (1), we have

$$T \frac{d^2 w}{dr^2} = \varepsilon_f \rho_f E_e \qquad (2)$$

In this work, the electric intensity is expressed as $E_e = U / \sqrt{r^2 + z_1^2}$, where $z_1$ is the distance from the positive electrode to the glass surface and $U$ is the applied voltage. Combining Eq. (2) and the boundary

condition of $w|_{r=r_0} = (dw/dr)|_{r=r_0} = 0$, the deflection of film in the electro-capillary peeling

$$w(r) = k_1 r \left( \ln \frac{r}{r_0} - 1 \right) + k_1 r_0 \qquad (3)$$

where $k_1 = \varepsilon_f \rho_f U / T$ and $r_0$ is the peeling length at present. Due to the ultrasmall applied voltage and the electrode located far away from the film, we consider that the voltage $U$ on the film would degenerate into the zeta potential $\zeta$ with evolution, and $k_1$ would become $k_2 = \varepsilon_f \rho_f \zeta / T$. Therefore, the lifting height does not evidently vary when the voltage changes from 2.5 to 4.5 V (corresponding to Fig. 2d).

Considering that the peeling process of the film on the rigid substrate is similar to the fracture. In this process, the Maxwell stress competes with the film's tension stress and adhesion stress to achieve film detachment. The adhesion work of the bonding layer is expressed as $U_a = f_{adhesion} r_0 l_0$, where $f_{adhesion}$ is the wet adhesion between the film and substrate. In addition, a state of plane strain is assumed to be

present exists on the film, and the elastic energy is described as $U_e = E\varepsilon r_0 l_0 d_0$. At the onset of electro-capillary peeling, the liquid cannot wet the bonding layer due to the large Laplace pressure (Fig. 4b). Under these conditions, the action length of the electric force is the width of induced charge aggregation. Therefore, the work of the Maxwell stress is described by $U_p = F^M z_0 = T^M l_0 \lambda_D k_1 r_0$, where $F^M$ is the electric force and $\lambda_D$ is the Debye length. According to the principle of energy minimization, we can obtain the critical electric force

$$F^M = \frac{1}{k_2}(E\varepsilon l_0 d_0 + f_{adhesion}l_0) \qquad (4)$$

After neglecting the term $O(f_{adhesion}l_0)$ and substituting $k_2 = \varepsilon_f \rho_f \zeta / T$ into Eq. (4), the critical peeling voltage becomes the following:

$$U_c = \frac{E\varepsilon d_0 r_0 f_{adhesion}}{\lambda_D \varepsilon_f^2 \rho_f^2 \zeta} \qquad (5)$$

We infer from Eq. (5) that the critical peeling voltage is directly proportional to the thickness and elastic modulus ($U_c \propto d_0$, $U_c \propto E$). The critical peeling voltage would be greater for harder or thicker films. The observations from Fig. 4c further demonstrate the deduction that the critical peeling voltage of electro-capillary peeling is proportional to the film thickness and elastic modulus, which is consistent with Eq. (5).

Substituting Eq. (3) and the net pressure into Poiseuille law, we obtain the liquid spreading rate

$$w^2 \frac{dp}{dr} = \mu v \qquad (6)$$

where $\mu$ and $v$ are the viscosity and spreading rate of liquid droplet, respectively. The relationship between film deflection and spreading length can be described by $w \sim r$. To determine the spreading law, we substitute the net pressure into Eq. (6), and the resulting relationship is that

$$r \sim \frac{\varepsilon_f^3 \rho_f^3 \zeta^2 U t}{\mu E^2 d_0^2} \qquad (7)$$

where $t$ is the peeling time. This resulting relationship indicates that the peeling length versus time is $r \sim t$, and the impact of the applied voltage, film thickness and elastic modulus on the peeling length are described by $r \sim U$, $r \sim d_0^{-2}$ and $r \sim E^{-2}$, respectively. Experimental observations of peeling length versus time, applied voltage, and film properties are shown in Fig. 4d, e. The solid line is the slope proposed in theory and discrete points are the peeling length observed in the experiment. The peeling time and applied voltage, the relationship between them and the peeling length is consistent with the analysis due to the bending stiffness's assumption having no effect. Within the uncertainties, an agreement is shown between the predicted and experimental data on the relationship between peeling length and peeling time/applied voltage, and the working mechanism describes the trend of the influence of thin film thickness and elastic modulus on the peeling length.

**Electro-capillary peeling with ultralow strain for functional device protection**

Due to long-term reliability being a major challenge for the commercialization of functional devices on films, developing a nondestructive method for peeling and transferring is significant in these applications. Different from the traditional approach, the electro-capillary peeling method has a unique detachment mode that detaches films from the substrate by exploiting the driving liquid to percolate and spread into the bonding layer. Characterized by using the three-dimensional digital image correlation (3D DIC, see Methods), the displacement and

strain fields are shown in Fig. 5a, b (Supplementary Movie 4). The film's displacement was only 0.152 mm in the $z$-direction and its strain was less than 0.00332 during the peeling process. Compared with the traditional method, the film detached by the electro-capillary peeling method had a much smaller strain than that peeled by the debonded strip and blister method (Fig. 5c). At the same peeling rate, the strain of film peeled by the blister approach was almost six times greater than that detached by the electro-capillary peeling method. Figure 5d shows the performance of ZnO nanorods on the film after detachment by using the electro-capillary peeling method and the debonded strip. Many cracks occurred on the ZnO nanorods layer when the film was peeled by using the debonded strip method ten times (Supplementary Fig. 6). Yet, the ZnO nanorods layer was adequately preserved in the electro-capillary peeling method. The size of the crack is about 2 μm in width and 10 μm in height. In addition, previous studies have shown that many micro/nanodevices on the film would be damaged due to the large film strain[41,42]. For this reason, the electro-capillary peeling method would be a valid approach to protect the functional device in these applications.

## Discussion

Above, we have provided experimental evidence of the electro-capillary peeling method that achieves thin film detachment by driving liquid to percolate and spread into the bonding layer under electric fields. The electro-capillary peeling method demonstrated that the electric field could control the liquid to peel off films from the substrate. Evaluated by various applied voltages, observation indicated that the peeling length $r$ was proportional to the applied voltage $U$ (corresponding to Fig. 2). It could be inferred from Eq. (7) that the peeling speed of electro-capillary peeling was expressed as $v \sim \varepsilon_f^3 \rho_f^3 \zeta^2 U / (\mu E^2 d_0^2)$. Based on the practical requirement, the peeling speed and detaching length could be precisely controlled by the applied voltage. Systematic experiments conducted with different films further showed our approach was suitable for a broad range of films although the detachment performance was affected by the film thickness, elastic modulus, and type (corresponding to Fig. 3). Additionally, the acquired deformation field showed that the film had an ultralow strain when it was detached by using the electro-capillary peeling method. For the long-term reliability of functional devices on the film, the electro-capillary peeling method would be a valid option for the detachment and transfer in various applications (corresponding to Fig. 5). Ultimately, the experimental setup and the critical peeling voltage indicated that the electro-capillary peeling method could easily peel the PDMS film (thickness of 100 μm and elastic modulus of 1.0 MPa) from the ITO glass by using a 0.7 V power supply (corresponding to Figs. 1, 4). This result also indicated that the films could be detached by using an AAA battery in practice, which would be highly appealing.

To evaluate the electro-capillary peeling method in long-term detaching, a strategy of replenishing liquid during the electro-capillary peeling was proposed for large-area film detachment (Supplementary Fig. 7). Exploiting this strategy, a thin film with a diameter of 5 cm was entirely peeled off at about 8.3 min (Supplementary Fig. 8). The detached thin film could be picked up and transferred to another substrate (Supplementary Fig. 8). Statistical results of peeling length revealed that the electro-capillary peeling method had a stable detaching rate in the long-term detachment when the electrolyte solution was adequate. In addition, we investigated the characterization of the substrate and thin film after electro-capillary peeling (Supplementary Figs. 9, 10). The photomicrograph and energy dispersive spectroscopy (EDS) results showed that some potassium chlorides (KCl) were deposited on the substrate but not on the film after peeling (Supplementary Fig. 9). The deposited potassium chlorides (KCl) could be cleaned off by water scrubbing. After transferring detached films onto other substrates, the thin film profile was consistent with the initial state and had no residual deformation (Supplementary Fig. 10).

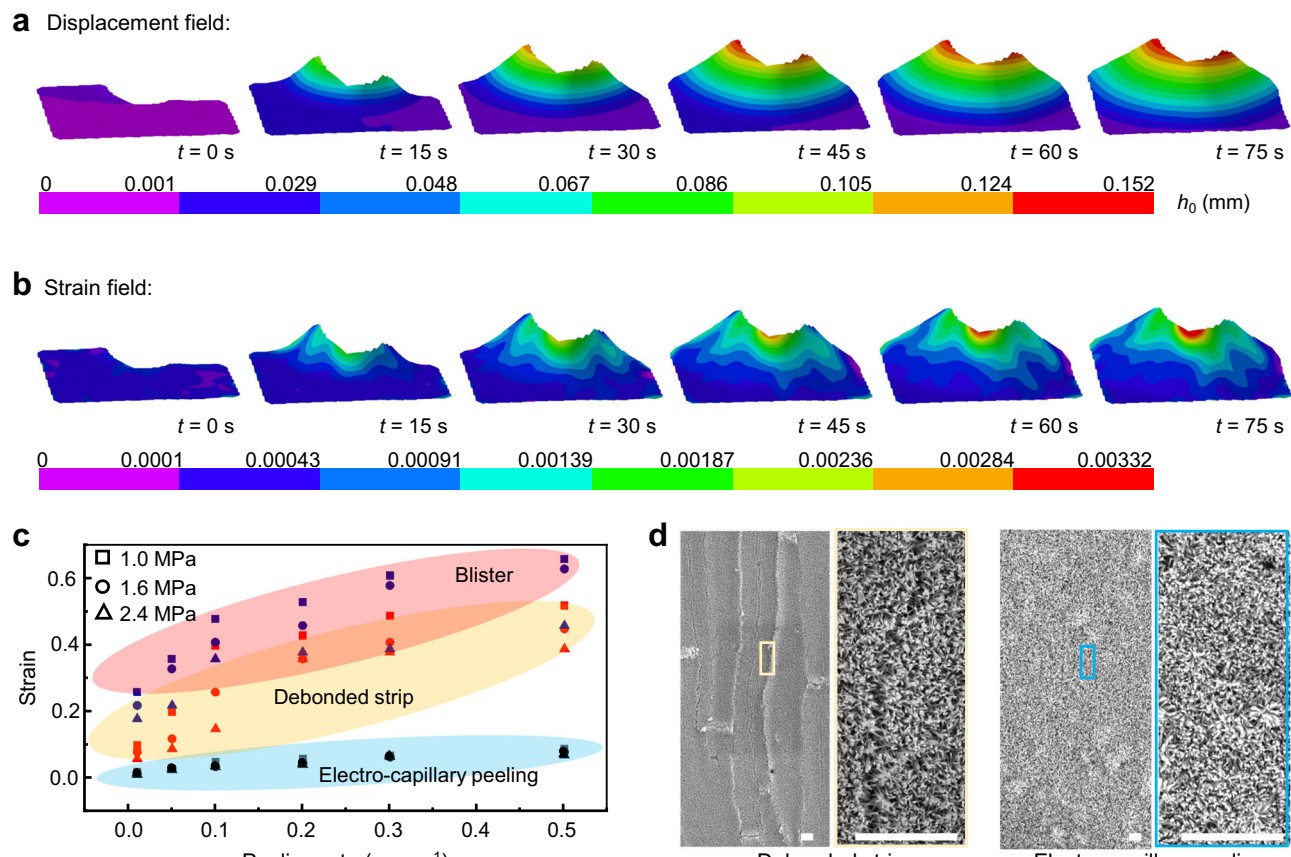

**Fig. 5 | Deformation of the film during the electro-capillary peeling process.** **a**, **b** Displacement and strain fields of the film detached using the electro-capillary peeling method. The displacement and strain fields exhibit deformation in the *z*-direction. Color bars represent the displacement and strain values. **c** Strain of film versus peeling rate in the various methods. The debonded strip method peels the film at an angle of 90°. The thickness of the tested films is 100 μm. **d** Characterization of the ZnO nanorods on the film after peeling ten times. The ZnO nanolayer occurs many cracks when a thin film is detached by the debonded strip but remains intact when peeled off by the electro-capillary peeling method. Scale bars are 10 μm.

Although many approaches have been proposed to detach thin films from the substrate[22–26], the electro-capillary peeling method is fundamentally different from them. First, the electro-capillary peeling had a distinct detaching mode that achieved thin film detachment by driving the liquid to percolate and spread into the bonding layer under electric fields. This mode could be performed by a DC power supply at any time and did not require a complicated chemical preparation. Compared with capillary peeling[29,30], the electro-capillary peeling method is an active control technique and appropriate for a broad range of films, including hydrophobic, hydrophilic, and even super hydrophilic films (corresponding to Fig. 3). The water blister and peeling test presented that the electro-capillary peeling method readily detaches the film of the wet adhesion from 37 to 3656 mJ m$^{-2}$ (Supplementary Tables 1–3). It was inferred from the working mechanism that the applied voltage could be increased to detach thin film with a strong bonding layer, indicating that the electro-capillary peeling method would be suitable for stronger interface detachment than the capillary peeling approach. In addition, the electro-capillary peeling method could be used in many solutions. Extended experimental observations performed in CaCl$_2$, CuSO$_4$, LiCl, NaCl, KOH, and NaOH solutions with concentrations of 0.1, 0.3, 0.5, and 1.0 mol L$^{-1}$ demonstrated that the electro-capillary peeling method could be applied in neutral, acidic, and alkaline solutions with a wide-range of concentrations (Supplementary Figs. 11, 12). For some thin films vulnerable to water, the electro-capillary peeling method could be applied to detach them using organic solvents, such as alcohol, acetone, and glycerol solutions (Supplementary Fig. 13).

Finally, the electro-capillary peeling method could also easily detach fully attached micro/nanofilms due to the special peeling mode. With the wide use of micro/nanofilms, our new method of "electro-capillary peeling" should attract much interest from academics and industry and potentially facilitate "water-based peeling" utilized in practical applications.

## Methods

### Characterization of the thin films electro-capillary peeling

In the electro-capillary peeling method, the experimental setup is schematically shown in Fig. 1. Before each test, the ITO glass (resistivity <6 Ω, and 10 cm in diameter) was rinsed in alcohol solution (99.97%) to remove the dust. After the PDMS film was treated by using the plasma method, a hole (2 mm in diameter) was fabricated at the center and adhered to the ITO glass surface. A micropump (Pump 11 Elite, Harvard Apparatus, USA) injected an electrolyte droplet (1.0 mol L$^{-1}$ KCl solution) with a volume of 20 μL into this prefabricated hole, and then an electric field was applied to the droplet by a DC supply power (LPS3020D, Lodestar, China). One of the Pt wire electrodes (99.99%, 200 μm in diameter) was inserted vertically into the liquid droplet and connected to the positive pole, and the other Pt wire electrode was fixed on the ITO glass surface and attached to the negative pole. Observation of the electro-capillary peeling was recorded by two cameras (JHUM1204s-E, Jinghang, China) mounted on an independent *XYZ* stage at 30 frames per second. The displacement of the droplet was obtained by using ImageJ software.

## Preparation of PDMS film

A kit including PDMS base and agent (Sylgard 184, Dow Corning) was used to prepare the PDMS films. The liquid PDMS mixture consisted of the base and curing agent with a weight ratio of 10:1 and was degassed and poured into a petri dish. A fixed volume of liquid PDMS mixture (8 mL) was deposited onto the center of the silicon wafer followed by a spin coating step. The PDMS film had a uniform thickness immediately after spin-coating for 30 s by a spin coater (WS-400B-6NPP-LITE, Laurell, USA). In this step, the thickness of the PDMS film was obtained with a designed rate. The PDMS films with thicknesses of 25, 50, 100, 200, and 300 μm were fabricated with spin rates of 3100.0, 1250.0, 650.0, 420.0, and 270.0 rpm s$^{-1}$, respectively. This PDMS film was cured on a levelled hot plate and finally transferred to a box furnace. After curing for 100 min at 65 °C, the PDMS film was ready for further experiments. Additionally, the PDMS films with elastic moduli of 1.0, 1.6, and 2.4 MPa were achieved with base and curing agent weight ratios of 20:1, 15:1, and 10:1, respectively. The other operations were the same as above.

## Fabrication of ZnO nanorods on PDMS films

To explore the electro-capillary peeling method in the protection of nanomaterials, ZnO nanorods were fabricated on a PDMS film. The crystal seed solution was prepared as follows: 5 g of Zn(Ac)$_2$·2H$_2$O, 0.8 g of monoethanol-amine (purchased from Shanghai Zhenpin Chemical Co., Ltd., China), and 20 mL of ethylene glycol monomethyl ether (purchased from Zhongshan Xinxin Chemical Co., Ltd., China) were mixed and stirred with a magnetic stirrer until the liquid became transparent. The growth liquid was achieved as follows: 0.4 g of hexamethylenetetramine (purchased from Beijing Lanyi Chemical Co., Ltd., China) and 0.82 g of Zn(NO$_3$)$_2$·6H$_2$O (purchased from Beijing Lanyi Chemical Co., Ltd., China) were mixed in 100 mL deionized water and stirred to become transparent. The PDMS surface was covered by the crystal seed via dip-coating and treated at 320 °C for 2 min. The ZnO nanoseed was achieved on the PDMS surface. The PDMS surface with the growth liquid was placed into the reaction kettle (100 mL) at 95 °C for 12 h. The ZnO nanorods were planted on the film (corresponding to Fig. 5d) and ready for further experiments.

## Preparation of electrolyte solution

In the electro-capillary peeling method, we selected the KCl solution as the electrolyte solution, due to its good electrical properties under electric fields. KCl solutions with concentrations of 0.1, 0.3, 0.5, and 1.0 mol L$^{-1}$ were prepared by dissolving their powders (Sigma Aldrich Shanghai Trading Co., Ltd., China) in deionized water at room temperature. In addition, to explore the performance of electro-capillary peeling in various electrolyte solutions, we further fabricated CaCl$_2$, CuSO$_4$, LiCl, NaCl, KOH, and NaOH solutions with concentrations of 0.1, 0.3, 0.5, and 1.0 mol L$^{-1}$. The pH values of these electrolyte solutions were measured with a pH meter (PHSJ-4, INESA Scientific Instrument Co., Ltd., China) and the average value of six measurements was calculated, which is shown in Supplementary Fig. 11.

## Electron microscopy and wettability characterization

The topographies of the bonding layer, ZnO nanorods, and film on the ITO glass surface were observed by environmental scanning electron microscopy (ESEM, Quanta FEG-250, FEI, USA) at a voltage of 10–15 kV. The chemical constitution of the substrate and thin film before and after peeling was characterized by a scanning electron microscope (SEM) with energy dispersive spectrometer (EDS) analysis (NOVA Nano SEM 430 + EDS, FEI, USA) and the size of cracks on the ZnO nanolayer was tested by an atomic force microscope (AFM, Multimode Pico-Force, Veeco metrology inc, USA). The contact angle of a liquid droplet with a volume of 10 μL on the film and ITO glass surface was measured

by a video-based contact angle measuring device (DSA100, KRÜSS Scientific, Germany) with a precision of ± 0.1°.

## Water blister test of the bonding layer wet adhesion

Devices were mounted with the tested films (including PDMS, PET, and PEN) facing up on an independent *XYZ* stage to ensure the alignment of the device with a horizontal microscope. The inlet of the microchannel on the tested films was connected to a syringe, which was mounted on a micropump (Pump 11 Elite, Harvard Apparatus, USA). Dyed water was prepared with DI water dissolved by a red dye and was injected through the inlet underneath the tested film with a flow rate of 0.16 mL min$^{-1}$. The schematic of the experimental setup used for the water blister test is shown in Supplementary Fig. 2. Two cameras were used to capture the water blister test process. Once the system achieved equilibrium, especially by inspecting that the debonded radius did not further develop, the blister's radius and height were measured by top-view and side-view visualization of the blister, respectively.

The energy release rate $G$ as a function of the blister height $h$, we use expressions provided by Sofla et al.[37].

$$G = \frac{\bar{E}d_0^5}{2a^4}(12\bar{M}(h)^2 + \bar{N}(h)^2) \tag{8}$$

where $\bar{M}(h)$ and $\bar{N}(h)$ are the dimensionless moment per unit length and the normal force, respectively, and $\bar{E} = E/(1 - \nu^2)$. The expressions for $\bar{M}(h)$ and $\bar{N}(h)$ are given as

$$\begin{cases} \bar{M}(h) = \frac{2}{3}\bar{h} + \left(\frac{m(\nu)\bar{h}^{1.25}}{2.2+\bar{h}^{1.25}}\right)\bar{h}^2, \\ \bar{N}(h) = n(\nu)\left[-0.255\bar{h}^2 \exp\left(-0.16\bar{h}^{1.3}\right) + 0.667\bar{h}^2\right] \end{cases} \tag{9}$$

where the dimensionless deflection is $\bar{h} = h/d_0$ and

$$\begin{cases} m(\nu) = 0.509 + 0.221\nu - 0.263\nu^2 \\ n(\nu) = 0.809 + 1.073\nu - 0.816\nu^2 \end{cases} \tag{10}$$

The Poisson's ratio $\nu$ used is 0.35. The wet adhesion measurements are provided in Supplementary Tables 1–3.

## Characterization of the thin film profile

The thin film profile before and after peeling was collected using a probe step profiler (Dektak XT, Bruker, Germany) with a 10 μN force on the microneedle (12.5 μm in diameter). The tested thin films had a diameter of 30 mm and a thickness from 25 μm to 300 μm. As shown in Supplementary Fig. 10, two profile parameters were tested by the probe step profiler, including the contour lines of the inner and outer circle boundary in the *xoy* plane and the *z*-coordinate of the top surface. The *z*-coordinate of the top surface was collected from 4 scan paths.

## Mechanical test of the film

1) Peeling test for films on the ITO glass surface. We performed peeling experiments of films on a tensile test apparatus equipped with a 90° peeling device and a 10 N load cell. The peeling rate was controlled with a stepper motor with a precision of 0.001 mm s$^{-1}$. 2) Measurement of Young's modulus. Rectangular strips were cut from the films (PDMS, Hydrogel, PET, and PEN) with dimensions of 80 mm × 20 mm × 1 mm. These rectangular strips were tested on a tensile test apparatus (All Around, Zwick Roell, Germany) equipped with a 100 N load cell. Samples were stretched at room temperature (20 °C) with no noticeable residual deformation and negligible hysteresis.

## Displacement and strain field of the film

Three-dimensional digital image correlation (3D DIC) with a spatial resolution of 10 μm was used to characterize the film's displacement and strain field in the electro-capillary peeling method (Supplementary Fig. 14). A micron particle was applied to track the positional relationship of the film in the 3D DIC. The acquired image was observed by two high-speed cameras and analyzed in the VIC-3D 9 system to characterize the film's deformation (Supplementary Fig. 14 and Supplementary Movie 4). The corresponding displacement and strain field are shown in Fig. 5a, b.

## Data availability

The data that support the findings of this study are available from the corresponding author upon request. The data generated in this study are provided in the Source Data file. Source data are provided with this paper.

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

## Acknowledgements
This work was jointly supported by the National Key R&D Program of China (Grant No. 2022YFA1203200, Y.-P. Z.) and the National Natural Science Foundation of China (Grant Nos. 12241205, 12032019, Y.-P. Z.).

## Author contributions
Y.-P.Z. conceptualized the idea and supervised the research. P.L. and X.H. conceived the idea and designed the experiment. P.L. collected the datasets and drafted the manuscript. All authors read, contributed to the discussion, and approved the final manuscript.

## Competing interests
The authors declare no competing interests.
