## [Peer Review File · Nature Communications]

REVIEWER COMMENTS

Reviewer #2 (Remarks to the Author):

In this manuscript, the authors presented an approach for detachment of thin films. This topic has garnered the attention of several applications, as discussed in the manuscript. The authors conducted carefully designed experiments to illustrate the effectiveness of their approach, i.e. the electro-capillary peeling of thin films. They argued that their method is a "simple physical peeling method" in contrast to other approaches involving modifications of the bonding layer's properties (I do not necessarily agree with this statement). As such, they suggest that their methodology "is perhaps an innovative option" (line 59).

Although I found the experimental results interesting and insightful, I do not think this manuscript is suitable for publication in Nature Communications because it does not offer broad and major new insights that could justify its publication in this journal. In my opinion, this analysis does not meet the high standard expected from the papers published in Nat. Commun. I think this analysis should be published in more specialized journals (e.g. Soft Matter could be an ideal home for such an analysis).

Furthermore, this manuscript certainly requires further proofreading to improve its quality. At some points it was difficult to understand the language and the key message of the sentences. Here are only a few examples where the language can be improved (too many to list all):

- Line 25, 26: "Planted by designed structures and devices, thin films are achieved to a specific function". I'm not sure what the authors are trying to say here.

- Line 32: "method for thin films is quietly attractive". I do not understand this sentence.

- Line 30: "such multilayered thin film systems are often prone to interfacial delamination/crack under various loading conditions". Something like "formation" or "initiation" after "crack" is missing or you could replace "crack" by "cracking".

- Line 33: "With the rapid development of miniaturized components, the thickness of film becomes micro/nanoscale". I kind of can guess what they mean here, but it needs revision.

-Line 70: "For the advantages, the electro-capillary peeling method is of significance for the promising application of thin film's detaching and transferring". Difficult to read.

As such, in my opinion the manuscript should go through professional proofreading. That way the reviewers will not be distracted by grammar issues and will be able to focus on the scientific contribution of the work and its novelty.

Reviewer #3 (Remarks to the Author):

Review Report: NCOMMS-23-07640

Electro-capillary peeling of thin films

The paper discussed the use of electrowetting to improve the control of capillary peeling. The authors tested multiple films with controlled thickness and elastic modulus. The propagation speed of the peeling front was also measured to validate the authors' proposed theoretical framework. The experimental results agreed nicely with the scaling law. The authors also demonstrated the technique's ability to peel off films with different adhesion, wettability, and elasticity.

Overall, the manuscript presented a new and interesting new direction in thin film processing. The science of the work is rigorous and thorough. I can also see direct and immediate societal impacts. However, I also have some major concerns that the authors should address before the paper can be considered to be published at Nature Communications.

Major comments:

1) I think the authors made a fundamental mistake on the concept of "adhesion" during capillary peeling. All the adhesion estimations and measurements are performed in a dry state, while the effective adhesion during capillary peeling is supposed to be "wet adhesion", or "under-water adhesion" (See main text reference [30]). The authors characterized adhesion using peeling tests in the air and estimate the dry adhesion to be about 10 J/m². This number is, in fact, much smaller than the scale of PDMS wet adhesion on glass (only 0.2 J/m² as measured by water blister test, see [1]). I would encourage the authors to repeat the adhesion measurement experiments in water and

revise the manuscript accordingly to emphasize the role of wet adhesion. As the actual adhesion is likely weak, the claim that capillary peeling can peel off “strong interfaces” is probably not valid (depends how different the wet and dry adhesion is for the interface of interest). More in-depth discussion on the applications is necessary.

2) The manuscript mostly describes the initial stage of the peeling process, where the peeling distance grows linearly with time. However, the growth slows down after 1 minute for all the data shown in the figure. The authors should at least provide one set of long-term peeling data (maybe 10 min?) with brief discussion, so that the readers can have a more complete picture of the peeling process.

3) Despite the ultimate goal of the work is to peel off an entire film, the authors only demonstrated the initiation of the peeling process with a droplet. It is doubtful whether a single droplet can peel-off a large film (unlikely). I guess many readers would also be curious to see the complete peeling of a film, so I encourage the authors to demonstrate the peeling off a 5 cm * 5 cm (or larger) sized film in the revised manuscript if possible.

4) The authors should pay more attention to the language used and avoid jargon (For example, “universal functional materials (line 4)”), grammar mistakes, meaningless terms (For example, “it is excitingly observed that (line 86)”), and novelty claims (For example, “we present an innovative detaching approach (line 7)”).

Additional comments:

1) The authors should specify the interface being referred to when discussing the ultra-low voltage of only 0.7 V. The voltage is not even needed to initiate capillary peeling for many weak interfaces (such as polystyrene-glass interface) as the authors pointed out themselves by citations.

2) The authors should give more specific definition of the characteristic length ($\sim 10^{-4}$ m) that is mentioned in line 166.

3) The authors should briefly discuss the high local-strain instability and fingering shown in Figure 2b and how it affects the overall thin film transfer performance.

4) The authors should explain why Figure 2e stops at 80s.

5) Despite the experimental data correspond pretty well with the scaling law, I still encourage the authors to use the experimental data to determine the term “ f_{adhesion} ” and discuss whether it is a consistent value over all the samples being used. It is supposed to have a constant value for a specific interface regardless of the coating thickness and coating elastic modulus. If not, the authors should discuss the discrepancy.

References

[1] Sofla, A., Seker, E., Landers, J. P., & Begley, M. R. (2010). PDMS-glass interface adhesion energy determined via comprehensive solutions for thin film bulge/blister tests. *Journal of Applied Mechanics*, 77(3).

Reviewer #4 (Remarks to the Author):

The manuscript introduces a novel method for the detachment of thin films from substrates called the electro-capillary peeling method. Unlike traditional techniques, this method uses an electric field to drive a liquid into the bonding layer, allowing the film to detach. This is the first example of using an electric field for this purpose. This method is effective across a wide variety of films and voltages, showing that it is an active control technique. The detachment process is driven by Maxwell stress in the electric field, which competes with the film's adhesive and tension stresses. The authors also provide mathematical relationships for peeling length with respect to time, applied voltage, film thickness, and elastic modulus. The film's deformation under this method showed a significantly lower strain, making it suitable for protecting functional devices. The study further reveals that a relatively low critical peeling voltage can be used to detach films across various solutions of different concentrations. Overall, this paper is of high quality, and I recommend the publication of this paper after addressing the following points.

- Some materials are vulnerable to water, which can induce the oxidation or dissolution of some films. I am curious if other organic solvents can also be used.

- The application of voltage on the water droplet can facilitate the electrochemical corrosion of the film. A high voltage would be needed for tightly bonded films to initiate detachment. Also, some films will be dissolved into water, and then the dissolved ions can be deposited somewhere on the substrate. I wonder if there are any additional test results regarding this point.

- As the detachment proceeds, the distance between the Pt wire and the very interface where the detachment occurs will change. So, I believe a more dynamic system should be developed to maintain a constant detachment speed.

- More demonstration of detached films is needed. I expect there will be deformation of detached films after transferring them onto other substrates.

- In order to verify the integrity of the detached thin films, additional device test results would be needed. (Data in Figure 5d are not sufficient.)

August 5, 2023

General Description for Reviewers

I, along with my coauthors, would like to submit the revised manuscript entitled “**Electro-capillary peeling of thin films**” for publication in *Nature Communications*.

Manuscript ID: NCOMMS-23-07640

Title: Electro-capillary peeling of thin films

We thank the reviewers for the time spent reading our manuscript, the recognition of our work, and the constructive comments and suggestions, which are very helpful in improving our paper. After carefully reading the reviewers’ comments, we diligently made extensive efforts to address the reviewers’ queries in the revised manuscript. These revisions made great improvements to our manuscript that will be suitable for publishing in *Nature Communications*.

With my best regards.

Sincerely yours,

Professor Ya-Pu Zhao (corresponding author)

State Key Laboratory of Nonlinear Mechanics

Institute of Mechanics, Chinese Academy of Sciences

Beijing 100190, CHINA

Phone: 0086-10-82543932

E-mail: yzhao@imech.ac.cn

Authors' Responses to Reviewer #2

In this manuscript, the authors presented an approach for detachment of thin films. This topic has garnered the attention of several applications, as discussed in the manuscript. The authors conducted carefully designed experiments to illustrate the effectiveness of their approach, i.e., the electro-capillary peeling of thin films. They argued that their method is a "simple physical peeling method" in contrast to other approaches involving modifications of the bonding layer's properties (I do not necessarily agree with this statement). As such, they suggest that their methodology "is perhaps an innovative option" (line 59).

Response: We thank the referee for the time spent on our paper and the constructive comments. We are pleased that the referee found our experimental results “interesting and insightful”. Empowered by the referee’s concern about our work’s novelty, broad and major new insights, and language polishing, we carefully made extensive efforts in the revised manuscript to address these queries. The details of the changes that we made are shown in the following response, and the revisions of the text are highlighted in the text body.

Regarding the assertion of a simple physical peeling method.

We appreciate the referee’s insightful comments. In our original manuscript, we demonstrated that the electro-capillary peeling method achieved thin film detachment by driving the liquid to percolate and spread into the bonding layer under electric fields. This peeling method could be performed by a DC supply power at any time and did not require a complicated chemical preparation. Therefore, we suggested that the electro-capillary peeling approach was a “simple physical peeling method”.

In the revised manuscript, we modified this problem by providing an appropriate description to illustrate our approach’s underlying working principles. The revisions in the revised manuscript are as follows:

1. In the Discussion section (Lines 288-291): “**First, the electro-capillary peeling had a distinct detaching mode that achieved thin film detachment by driving the liquid**

to percolate and spread into the bonding layer under electric fields. This mode could be performed by a DC power supply at any time and did not require a complicated chemical preparation.”.

Regarding the novelty and originality of the electro-capillary peeling method.

We thank the referee’s constructive comments. The claim that our method is “an innovative option” is not exactly concluded from the above assertion that our method is a “simple physical peeling method”. To show the novelty and originality of the electro-capillary peeling method clearly, we provided a detailed summary of previous studies related to our work and highlighted their differences from our research as well as the novelty and new contributions of this study (Figure R1).

1) The detachment mode of electro-capillary peeling is fundamentally different from traditional approaches. Electro-capillary peeling achieves thin film detachment by driving liquid to percolate and spread into the bonding layer under electric fields. This peeling mode fundamentally differs from traditional methods that generally achieve thin film detachment by applying tensile stress on the films to overcome the bonding layer adhesion. The detachment mode of the electro-capillary peeling method avoids a large film deformation during the detaching process. **For the long-term reliability of functional devices on the film, the electro-capillary peeling method would be a valid option for film detachment and transfer in various applications.** Meantime, the electro-capillary peeling mode does not require a clamping area or a substrate’s prefabricated hole, which is **especially suitable for the fully-attached micro/nanofilm detachment.**

2) The working mechanism of the electro-capillary peeling method differs from the previous studies. Theoretically, the electro-capillary peeling is achieved by utilizing the Maxwell stress to compete with the film’s adhesion stress and tension stress. Because the Maxwell stress is regulated by the applied voltage, the detaching speed/length of the electro-capillary peeling can be controlled by electric fields. **Based on the practical requirement, the peeling speed and detaching length of**

the electro-capillary peeling could be precisely controlled by the applied voltage. In addition, the working mechanism demonstrates that the applied voltage can be increased to detach the thin film with a strong bonding layer, indicating the electro-capillary would be suitable for detaching various thin films.

3) The electro-capillary peeling method is achieved by using a DC power supply. The electro-capillary peeling method achieves thin film detachment by using a DC power supply at any time and does not require a complicated chemical preparation. **In the experiment, the electro-capillary peeling easily detaches thin film at the voltage of 0.7 V.** With the wide use of micro/nanofilms, our method of “electro-capillary peeling” would be utilized in practical applications by using an AAA battery, which is highly attractive.

In the revised manuscript, we added a detailed description of our work’s novelty and contribution in the Abstract, Introduction, and Discussion sections. We believe that the interesting phenomena and the physics of the electro-capillary peeling method are novel enough and exhibit a broad impact in multiple areas to be suitable for publication in *Nature Communications*.

Traditional Method	Chemical strategy	Capillary peeling	Electro-capillary peeling
Large clamping area/ Pre-fabricated hole	Complicated chemical modification	Stepping motor/ Liquid	DC power supply/ Liquid
Tensile stress overcomes the adhesion	Reduce adhesion/ Dissolve bonding layer	Liquid penetrates	Electric field drives the liquid to penetrate
All types (Macroscale)	All types	Hydrophobic (Micro/nanoscale)	All types (Micro/nanoscale)
Large	N/A	Small	Small
Needed	Working principle	Thin film	Film deformation

Figure R1. Comparison of our work with previous studies in thin films’ detachment.

Although I found the experimental results interesting and insightful, I do not think this manuscript is suitable for publication in Nature Communications because it does not offer broad and major new insights that could justify its publication in this journal. In my opinion, this analysis does not meet the high standard expected from the papers published in Nat. Commun. I think this analysis should be published in more specialized journals (e.g., Soft Matter could be an ideal home for such an

analysis).

Response: We thank the referee for the constructive comments and the recognition of the experimental results. According to the referee's insightful comment, we made extensive efforts to offer broad and major new insights about the electro-capillary peeling method in the revised manuscript, including:

- 1) Make a more profound analysis to demonstrate the novelty and contribution of the electro-capillary peeling method clearly, highlighting our work's strengths and broad interests.
- 2) Perform a detailed analysis of the electro-capillary peeling working mechanism to guide practical applications.
- 3) Conduct additional experiments to explore broader applications of the electro-capillary peeling method.

The details of the changes that we made are as follows.

1) Make a more profound analysis to demonstrate the novelty and contribution of the electro-capillary peeling method clearly, highlighting our work's strengths and broad interests.

In the revised manuscript, we made a more profound analysis to demonstrate the novelty and contribution of the electro-capillary peeling method clearly, highlighting this work's strengths and broad interests (Figure R1). The thorough analysis illustrates that **the electro-capillary peeling method involves a new peeling mode, working mechanism, and implementation device, which has not been reported in previous studies.** In addition, the electro-capillary peeling method exhibits great advantages in the micro/nanofilm detachment and transfer, including **small film deformation and ultralow applied voltage**, which would attract much interest from academics and industry. The brief description is as follows (detailed description as discussed in our response to Comment 1). First, the electro-capillary peeling mode fundamentally differs from traditional detaching approaches, achieving the thin film's detachment by driving liquid to percolate and spread into the bonding layer under electric fields. This peeling mode **avoids a large film deformation during the peeling process**, which is highly attractive for the long-term reliability of functional devices on the film.

Meantime, the electro-capillary peeling mode does not require a clamping area or a substrate's prefabricated hole, which is **especially suitable for the fully-attached micro/nanofilm detachment**. Second, the working mechanism of electro-capillary peeling differs from the previous studies. It is demonstrated that the electro-capillary peeling is achieved by utilizing the Maxwell stress to compete with the film's adhesion stress and tension stress. Based on the practical requirement, **the peeling speed and detaching length can be actively controlled by the applied electric fields to suit various applications**. Third, the electro-capillary peeling method can be achieved by a DC power supply at a voltage of 0.7 V. In practice, the electro-capillary peeling method **could be performed by using an AAA battery**.

2) **Perform a more detailed analysis of the electro-capillary peeling working mechanism to guide practical applications.**

In our original manuscript, we proposed a theoretical framework to explain the electro-capillary peeling method and the impact of the applied voltage and film properties on detachment behavior. In the revised manuscript, we added a more detailed discussion of the working mechanism to illustrate the detaching length/rate actively controlled by the applied voltages, which could be applied to guide practical applications. The detailed discussion is as follows.

i) **Add a detailed discussion to illustrate the detaching length/speed of the electro-capillary peeling actively controlled in practical applications.**

In the original manuscript, we demonstrated that the peeling length versus time is $r \sim t$, and the impact of the applied voltage, thickness, and elastic modulus on the peeling length are described by $r \sim U$, $r \sim d_0^{-1/2}$, and $r \sim E^{-1/2}$, respectively. It can be inferred from the above analysis that the peeling speed is proportional to the applied voltage U . In addition, the electro-capillary peeling can be stopped at any time by turning off the power. Therefore, the peeling speed and detaching length can be precisely controlled by altering the applied voltage in applications. The revision in the revised manuscript is as follows.

1. In the Discussion section (Lines 262-265): **“Evaluated by various applied voltages,**

observation indicated that the peeling length r was proportional to the applied voltage U (corresponding to Fig. 2). It could be inferred from Equation (7) that the peeling speed of electro-capillary peeling was expressed as $v \sim \varepsilon_f^3 \rho_f^3 \zeta^2 U / (\mu E^2 d_0^2)$. Based on the practical requirement, the peeling speed and detaching length could be precisely controlled by the applied voltage.”.

3) Conduct additional experiments to explore broader applications of the electro-capillary peeling method.

In practice, the electro-capillary peeling method would be applied in a long-term detachment or utilized to detach some materials vulnerable to water. Based on these practical requirements, we conducted additional experiments to show a more complete picture of the long-term peeling process and investigate organic solvents’ application in the electro-capillary peeling method. The observation showed that **the electro-capillary peeling method had a stable detachment behavior in long-term detaching and could be applied in water/organic solvents.** Therefore, the electro-capillary peeling method would be suitable for large-area film detachment and could be used to detach some materials vulnerable to water. The revision in the revised manuscript is as follows.

i) Long-term detachment behavior of the electro-capillary peeling. In the revised manuscript, we proposed a long-term detachment strategy by replenishing liquid during the electro-capillary peeling. Exploiting this strategy, we conducted long-time electro-capillary peeling experiments with thin films of 1.0, 1.6, and 2.4 MPa to characterize the long-term detaching behavior (Supplementary Fig. 7). The statistical results of peeling length indicated that the electro-capillary peeling method had a stable detaching rate in the long-term detachment when the electrolyte solution was adequate. For this reason, the electro-capillary peeling method would be a valid approach for large-area film detachment.

1. Supplementary Fig. 7 in Supplementary file:

Supplementary Fig. 7. A long-term detachment strategy of the electro-capillary peeling method. **a** Schematic illustration of the long-term detachment strategy. A micropump is used to replenish fluid during the electro-capillary peeling. **b** Side and top views of the long-term peeling process. Black, red, and blue boxes are used to mark the thin film detachment with elastic moduli of 1.0, 1.6, and 2.4 MPa, respectively. The tested time is 660 s, and the scale bars in the side and top views are 1 cm. **c** Statistical results of peeling length in the long-term peeling. The thickness of tested films in **b** and **c** are 100 μm , and the error bars in **c** are the standard deviation of the raw data.

2. In the Discussion section (Lines 275-280): “To evaluate the electro-capillary peeling method in long-term detaching, a strategy of replenishing liquid during the electro-capillary peeling was proposed for large-area film detachment (Supplementary Fig. 7). Exploiting this strategy, a thin film with a diameter of 5 cm was entirely peeled off at about 8.3 min (Supplementary Fig. 8). The detached thin film could be picked up and transferred to another substrate (Supplementary

Fig. 8). Statistical results of peeling length revealed that the electro-capillary peeling method had a stable detaching rate in the long-term detachment when the electrolyte solution was adequate.”.

ii) The organic solvents’ application in the electro-capillary peeling method.

Investigating the organic solvents’ application in the electro-capillary peeling method is greatly important to detach some thin films vulnerable to water. In the revised manuscript, we performed additional experiments to evaluate the applications of organic solvents in the electro-capillary peeling method, including ethanol, acetone, and glycerol solutions (Supplementary Fig. 13). The acquired experimental results showed that the organic solvents could be applied in the electro-capillary peeling method. For some thin films vulnerable to water, organic solvents could be applied to detach them via the electro-capillary peeling method. The revision is as follows.

1. Supplementary Fig. 13 in Supplementary file:

Supplementary Fig. 13. Electro-capillary peeling method using different types of organic solvents. a Schematic diagram of electro-capillary peeling applied with organic solvents. **b** The detaching process of the electro-capillary peeling method applied with organic solvents. The tested organic solvents are

(±)-Camphor-10-sulfonic acid/alcohol, (±)-Camphor-10-sulfonic acid/acetone, and sodium hydroxide/glycerol solutions. The concentrations of solutions are 0.5 mol L⁻¹, and the scale bars are 2 mm.

2. In the Discussion section (Lines 301-303): “For some thin films vulnerable to water, the electro-capillary peeling method could be applied to detach them using organic solvents, such as alcohol, acetone, and glycerol solutions (Supplementary Fig. 13).”.

The above results and analyses were added to the revised manuscript’s main text and supplementary file. These revisions showed that the electro-capillary peeling method has a broad range of applications and exhibited great advantage in the micro/nanofilm detachment, which provides profound and original insights into thin film detachment and transfer. We firmly believe that our work indeed shows significant scientific progress and a novel conceptual advance to match the standards of *Nature Communications*.

Furthermore, this manuscript certainly requires further proofreading to improve its quality. At some points it was difficult to understand the language and the key message of the sentences. Here are only a few examples where the language can be improved (too many to list all):

- Line 25, 26: "Planted by designed structures and devices, thin films are achieved to a specific function". I'm not sure what the authors are trying to say here.
- Line 32: "method for thin films is quietly attractive". I do not understand this sentence.
- Line 30: "such multilayered thin film systems are often prone to interfacial delamination/crack under various loading conditions". Something like "formation" or "initiation" after "crack" is missing or you could replace "crack" by "cracking".
- Line 33: "With the rapid development of miniaturized components, the thickness of film becomes micro/nanoscale". I kind of can guess what they mean here, but it needs revision.
- Line 70: "For the advantages, the electro-capillary peeling method is of significance

for the promising application of thin film's detaching and transferring". Difficult to read.

As such, in my opinion the manuscript should go through professional proofreading. That way the reviewers will not be distracted by grammar issues and will be able to focus on the scientific contribution of the work and its novelty.

Response: We thank the referee for the careful comments. In the revised manuscript, we made a thorough stylistic and grammatical polish of the full text to clearly convey the message, methods, and arguments. The details of the changes that we made are shown in the following.

1. Line 25, 26: "Planted by designed structures and devices, thin films are achieved to a specific function" is revised as "In general, thin films serve as base materials implanted by some functional structures and devices to achieve a specific ability."
2. Line 32: "method for thin films is quietly attractive" is revised as "method for thin films is highly attractive".
3. Line 30: "such multilayered thin film systems are often prone to interfacial delamination/crack under various loading conditions" is revised as "these multilayered thin film systems are often prone to interfacial delamination/cracking under various loading conditions".
4. Line 33: "With the rapid development of miniaturized components, the thickness of film becomes micro/nanoscale" is revised as "To achieve the miniaturization of the components, the thickness of the thin film used has increasingly become smaller and is on the micro/nanoscale level."
5. Line 70: "For the advantages, the electro-capillary peeling method is of significance for the promising application of thin film's detaching and transferring" is revised as "Therefore, the electro-capillary peeling method has promising applications in detaching and transferring thin films."
6. In addition, we engaged the services of Springer Nature Author Services (SNAS) to edit our paper, focusing on enhancing grammar, phrasing, and punctuation to improve the flow and readability of the text. The relevant certificate of language proofreading can be verified on the SNAS website using the verification code

5C68-36F7-1E96-5783-26CP. More details of the changes that we made are highlighted in the main text and Supplementary Information.

We thank the referee again for the kind help, constructive comments, and insightful suggestions. We are lucky to have the referee who provided very insightful and constructive comments and suggestions that fundamentally improved the scientific depth of our work. We believe that with the referee's help, the revised manuscript is suitable for publication in *Nature Communications* now.

Authors' Responses to Reviewer #3

The paper discussed the use of electrowetting to improve the control of capillary peeling. The authors tested multiple films with controlled thickness and elastic modulus. The propagation speed of the peeling front was also measured to validate the authors' proposed theoretical framework. The experimental results agreed nicely with the scaling law. The authors also demonstrated the technique's ability to peel off films with different adhesion, wettability, and elasticity.

Overall, the manuscript presented a new and interesting new direction in thin film processing. The science of the work is rigorous and thorough. I can also see direct and immediate societal impacts. However, I also have some major concerns that the authors should address before the paper can be considered to be published at Nature Communications.

Response: We thank the referee for the time spent on our paper and the constructive comments. We are pleased that the referee considers our work to present “a new and interesting new direction in thin film processing” and to be “rigorous and thorough”. Empowered by the referee's concern about the wet adhesion of the bonding layer, the characteristic of detaching speed in long-term detachment, and the entire electro-capillary peeling process for a large-area film, we conducted additional experiments and provided plenty of experimental results in the revised manuscript to address these queries. We believe that our paper's scientific merit is much improved after revising following the referee's comments. The details of the changes that we made are shown in the following response, and revisions of the text are highlighted in the text body.

Major comments:

1) I think the authors made a fundamental mistake on the concept of “adhesion” during capillary peeling. All the adhesion estimations and measurements are performed in a dry state, while the effective adhesion during capillary peeling is supposed to be "wet adhesion", or “under-water adhesion” (See main text reference

[30]). The authors characterized adhesion using peeling tests in the air and estimate the dry adhesion to be about 10 J/m^2 . This number is, in fact, much smaller than the scale of PDMS wet adhesion on glass (only 0.2 J/m^2 as measured by water blister test, see [1]). I would encourage the authors to repeat the adhesion measurement experiments in water and revise the manuscript accordingly to emphasize the role of wet adhesion. As the actual adhesion is likely weak, the claim that capillary peeling can peel off “strong interfaces” is probably not valid (depends how different the wet and dry adhesion is for the interface of interest). More in-depth discussion on the applications is necessary.

References

[1] Sofla, A., Seker, E., Landers, J. P., & Begley, M. R. (2010). PDMS-glass interface adhesion energy determined via comprehensive solutions for thin film bulge/blister tests. *Journal of Applied Mechanics*, 77(3).

Response: We thank the referee for this insightful comment. We noted that the characterized adhesion using the peeling test underwater would be more consistent with the effective adhesion in the electro-capillary peeling method. Following the referee’s constructive suggestions, we thus measured the wet adhesion of PDMS film on the ITO glass using the water blister test (see Methods section and Supplementary Fig. 2 in the revised manuscript). The measured wet adhesion is about 117 mJ/m^2 , which is consistent with previous literature [37]. In the revised manuscript, we updated the characterized adhesion of electro-capillary peeling by the wet adhesion of 117 mJ/m^2 .

Regarding the claim that electro-capillary peeling can peel off “strong interfaces”, we provided a detailed discussion to explain this issue in the revised manuscript. In this work, we used four types of thin films to evaluate the electro-capillary peeling method, including PDMS, PET, PEN, and hydrogel film. The wet adhesion of PDMS, PET, PEN, and hydrogel films on the ITO glass are about 117, 35, 66, and 3656 mJ/m^2 , respectively, which is much larger than the wet adhesion of the film used in the capillary peeling method (PS-Sapphire, PS-SiO₂, Teflon-SiO₂, and so on)

($< 30 \text{ mJ/m}^2$) [29, 30]. The wet adhesion of hydrogel films here was characterized by a peeling test underwater due to the hydrogel films being always destroyed in the water blister test. In the revised manuscript, we added a description to explain this problem. Additionally, the working mechanism established in our work demonstrated that the bonding layer adhesion versus the critical peeling voltage was $f_{\text{adhesion}} = U_c \lambda_D \epsilon_f^2 \rho_f^2 \zeta / (E \epsilon d_0 r_0)$. The applied voltage could be increased to detach the thin film with a strong bonding layer. Therefore, we suggest that the electro-capillary peeling method proposed in our work would be suitable for a strong interface detachment. In the revised manuscript, we added a detailed discussion to elaborate on this claim, avoiding misunderstandings among readers. The discussion in the revised manuscript is as follows.

1. Supplementary Fig. 2 in Supplementary file:

Supplementary Fig. 2. Water blister measurement of the bonding layer wet adhesion. **a** Schematic illustration of the water blister measurement. Dyed water is injected through the inlet underneath the thin film. The side-view and top-view images of the blister are captured to determine the maximum deflection h and

debonded diameter $2a$. **b** Representative image of the delamination front during a blistering test. The debonded radius (a) is estimated by fitting a curve to the edge of the region where the dyed fluid is visible. **c** Experimental results of aqueous injection. The debonded radius and deflection are tested with a thin film of various thicknesses, elastic moduli, and types. Debonded radius versus deflection to determine the interface wet adhesion. The error bars in **c** are the standard deviation of the raw data.

2. In the Results section (Lines 76-78): “The wettability showed that the bonded surface was hydrophilic, and the water blister test³⁷ (see Methods) presented that the wet adhesion of the bonding layer was approximately 117 mJ m^{-2} (Supplementary Fig. 2 and Supplementary Table 1).”.

3. In the Discussion section (Lines 294-298): “The water blister and peeling test presented that the electro-capillary peeling method readily detaches the film of the wet adhesion from 35 to 3656 mJ m^{-2} (Supplementary Tables 1 to 3). It was inferred from the working mechanism that the applied voltage could be increased to detach thin film with a strong bonding layer, indicating that the electro-capillary peeling method would be suitable for stronger interface detachment than the capillary peeling approach.”.

4. In the Methods section (Lines 359-375):

“Water blister test of the bonding layer wet adhesion. Devices were mounted with the tested films (including PDMS, PET, and PEN) facing up on an independent XYZ stage to ensure the alignment of the device with a horizontal microscope. The inlet of the microchannel on the tested films was connected to a syringe, which was mounted on a micropump (Pump 11 Elite, Harvard Apparatus, USA). Dyed water was prepared with DI water dissolved by a red dye and was injected through the inlet underneath the tested film with a flow rate of 0.16 mL min^{-1} . The schematic of the experimental setup used for the water blister test is shown in Supplementary Fig. 2. Two cameras were used to capture the water blister test process. Once the system achieved equilibrium, especially by inspecting that the debonded radius did not further develop, the blister’s radius and height were measured by top-view and side-view visualization

of the blister, respectively.

The energy release rate G as a function of the blister height h , we use expressions provided by Sofla et al.³⁷

$$G = \frac{\bar{E}d_0^5}{2a^4} \left(12\bar{M}(h)^2 + \bar{N}(h)^2 \right), \quad (8)$$

where $\bar{M}(h)$ and $\bar{N}(h)$ are the dimensionless moment per unit length and the normal force, respectively, and $\bar{E} = E/(1-\nu^2)$. The expressions for $\bar{M}(h)$ and $\bar{N}(h)$ are given as

$$\begin{cases} \bar{M}(h) = \frac{2}{3}\bar{h} + \left(\frac{m(\nu)\bar{h}^{1.25}}{2.2 + \bar{h}^{1.25}} \right) \bar{h}^2, \\ \bar{N}(h) = n(\nu) \left[-0.255\bar{h}^2 \exp(-0.16\bar{h}^{1.3}) + 0.667\bar{h}^2 \right], \end{cases} \quad (9)$$

where the dimensionless deflection is $\bar{h} = h/d_0$ and

$$\begin{cases} m(\nu) = 0.509 + 0.221\nu - 0.263\nu^2, \\ n(\nu) = 0.809 - 1.073\nu - 0.816\nu^2. \end{cases} \quad (10)$$

The Poisson's ratio ν used is 0.35. The wet adhesion measurements are provided in Supplementary Tables 1-3.”.

2) The manuscript mostly describes the initial stage of the peeling process, where the peeling distance grows linearly with time. However, the growth slows down after 1 minute for all the data shown in the figure. The authors should at least provide one set of long-term peeling data (maybe 10 min?) with brief discussion, so that the readers can have a more complete picture of the peeling process.

Response: We greatly appreciate the referee's constructive comments. In the revised manuscript, we proposed a long-term detachment strategy by replenishing liquid during the electro-capillary peeling. Exploiting this strategy, we conducted ten-minute electro-capillary peeling experiments with thin films of 1.0, 1.6, and 2.4 MPa to describe the long-term peeling behavior (Supplementary Fig. 7). Statistical results of peeling length showed that the electro-capillary peeling method had a stable detaching

speed in the long-term detachment when the electrolyte solution was adequate. The growth slightly slowed down after 1 minute, which was induced by a single droplet liquid level decreasing to affect the electric field. In the revised manuscript, we added a detailed discussion to explain this problem. The revision in the revised manuscript is as follows.

1. Supplementary Fig. 7 in Supplementary file:

Supplementary Fig. 7. A long-term detachment strategy of the electro-capillary peeling method. **a** Schematic illustration of the long-term detachment strategy. A micropump is used to replenish fluid during the electro-capillary peeling. **b** Side and top views of the long-term peeling process. Black, red, and blue boxes are used to mark the thin film detachment with elastic moduli of 1.0, 1.6, and 2.4 MPa, respectively. The tested time is 660 s, and the scale bars in the side and top views are 1 cm. **c** Statistical results of peeling length in the long-term peeling. The thickness of tested films in **b** and **c** are 100 μm , and the error bars in **c** are the standard deviation of

the raw data.

2. In the Discussion section (Lines 275-280): “To evaluate the electro-capillary peeling method in long-term detaching, a strategy of replenishing liquid during the electro-capillary peeling was proposed for large-area film detachment (Supplementary Fig. 7). Exploiting this strategy, a thin film with a diameter of 5 cm was entirely peeled off at about 8.3 min (Supplementary Fig. 8). The detached thin film could be picked up and transferred to another substrate (Supplementary Fig. 8). Statistical results of peeling length revealed that the electro-capillary peeling method had a stable detaching rate in the long-term detachment when the electrolyte solution was adequate.”.

3) Despite the ultimate goal of the work is to peel off an entire film, the authors only demonstrated the initiation of the peeling process with a droplet. It is doubtful whether a single droplet can peel-off a large film (unlikely). I guess many readers would also be curious to see the complete peeling of a film, so I encourage the authors to demonstrate the peeling off a 5 cm * 5 cm (or larger) sized film in the revised manuscript if possible.

Response: We thank the referee for the constructive suggestions. As discussed in our response to comment 2, we proposed a long-term detachment strategy by replenishing liquid during the electro-capillary peeling (Supplementary Fig. 7). Exploiting this strategy, we used a thin film with a diameter of 5 cm to conduct additional electro-capillary peeling experiments. The acquired experimental result showed that the electro-capillary peeling method had a stable detaching rate when the electrolyte solution was adequate, and the thin film with a diameter of 5 cm was entirely peeled off at about 8.3 min (Supplementary Figs. 7 and 8). In addition, we presented an example of thin film transfer in the electro-capillary peeling method. When the liquid spread in the bonding layer fully, the thin film was entirely peeled off and could be picked up for transfer to another substrate (Supplementary Fig. 8). In the revised manuscript, we added a detailed description to illustrate this process, and the revision is as follows.

1. Supplementary Fig. 8 in the Supplementary file:

Supplementary Fig. 8. The entire electro-capillary peeling process of a thin film (5 cm in diameter). a-b Side and top views of the entire electro-capillary peeling process. When the liquid spreads in the bonding layer fully, it seeps from the thin film's boundary. Yellow and red dot lines mark the thin film's boundary and the exudated liquid water, respectively. **c** Thin film transfer process. After the film is entirely detached by the electro-capillary peeling method, it can be picked up and transferred to another substrate.

2. In the Discussion section (Lines 275-280): “To evaluate the electro-capillary peeling method in long-term detaching, a strategy of replenishing liquid during the electro-capillary peeling was proposed for large-area film detachment (Supplementary Fig. 7). Exploiting this strategy, a thin film with a diameter of 5 cm was entirely peeled off at about 8.3 min (Supplementary Fig. 8). The detached thin film could be picked up and transferred to another substrate (Supplementary Fig. 8). Statistical results of peeling length revealed that the electro-capillary peeling method had a stable detaching rate in the long-term detachment when the electrolyte solution was adequate.”.

4) The authors should pay more attention to the language used and avoid jargon (For example, “universal functional materials (line 4)”), grammar mistakes, meaningless

terms (For example, “it is excitingly observed that (line 86)”), and novelty claims (For example, “we present an innovative detaching approach (line 7)”).

Response: We thank the referee for pointing out these problems. In the revised manuscript, we made a thorough stylistic and grammatical polish of the full text to avoid jargon, grammar mistakes, meaningless term, and novelty claims. Part of the changes we made are shown in the following.

1. Line 4: “Thin films are widely-used functional materials that have attracted much interest in academic and industrial applications.”.
2. Line 7: “Here, we develop an electro-capillary peeling strategy that achieves thin film detachment by driving liquid to percolate and spread into the bonding layer under electric fields”.
3. Line 86: “After the power was turned on, the liquid droplet percolated into the bonding layer and detached the PDMS film from the ITO glass (Fig. 1b and Supplementary Movie 1).”.
4. In addition, we engaged the services of Springer Nature Author Services (SNAS) to edit our paper, focusing on enhancing grammar, phrasing, and punctuation to improve the flow and readability of the text. The relevant certificate of language proofreading can be verified on the SNAS website using the verification code **5C68-36F7-1E96-5783-26CP**. More details of the changes that we made are highlighted in the main text and Supplementary Information.

Additional comments:

1) The authors should specify the interface being referred to when discussing the ultra-low voltage of only 0.7 V. The voltage is not even needed to initiate capillary peeling for many weak interfaces (such as polystyrene-glass interface) as the authors pointed out themselves by citations.

Response: We thank the referee for this helpful comment. In the revised manuscript, we added a detailed description to illustrate the interface being referred to a voltage of 0.7 V. The revision in the revised manuscript is as follows.

1. In the Results section (Lines 64-66): “In addition, observations of the critical

peeling voltage indicate that a polydimethylsiloxane (PDMS) film (thickness of 100 μm and elastic modulus of 1.0 MPa) is easily detached from indium tin oxide (ITO) glass by using the electro-capillary peeling method at a voltage of 0.7 V”.

2. In the legend of Figure 4c (Lines 226-228): “Critical peeling voltage versus thin film thickness and elastic modulus. The critical peeling voltage versus thickness and elastic modulus are tested with PDMS film of 1.0 MPa and 100 μm , respectively.”.
3. In the Discussion section (Lines 271-273): “Ultimately, the experimental setup and the critical peeling voltage indicated that the electro-capillary peeling method could easily peel the PDMS film (thickness of 100 μm and elastic modulus of 1.0 MPa) from the ITO glass by using a 0.7 V power supply (corresponding to Figs. 1 and 4).”.

2) The authors should give more specific definition of the characteristic length ($\sim 10^{-4}$ m) that is mentioned in line 166.

Response: We thank the referee for pointing out this problem. In the revised manuscript, we gave a specific definition of the characteristic length mentioned in the theoretical section. The revision in the revised manuscript is as follows.

1. Lines 165-167: “Because the working characteristic length of the liquid layer thickness (10^{-4} m) in this work is much smaller than that of the Bond number (10^{-3} m)³⁴, the effect of fluid gravity can be neglected (corresponding to Fig. 2d).”.

3) The authors should briefly discuss the high local-strain instability and fingering shown in Figure 2b and how it affects the overall thin film transfer performance.

Response: We thank the referee for the insightful comments and suggestions. The “fingering” graphics shown in Figure 2b were the pattern formed by the potassium chloride (KCl) crystallization, which was not the high local-strain instability. In the revised manuscript, we conducted photomicrograph and EDS experiments to characterize this graphic. The observation showed that some potassium chlorides

(KCl) were deposited on the substrate due to the increased solution concentration induced by liquid evaporation or electrolysis (Supplementary Fig. 9). After a water scrubbing, this pattern could be cleaned off. In addition, the observation showed that the thin film detachment had a similar detaching behavior at various voltages, which was not affected by this “fingering” pattern during the electro-capillary peeling. A detailed description of the “fingering” pattern was added in the revised manuscript.

1. In the legend of Figure 2b (Line 122): “The “fingering” graphics are the pattern formed by the potassium chloride (KCl) precipitation.”.
2. In the Discussion section (Lines 282-284): “The photomicrograph and energy dispersive spectroscopy (EDS) results showed that some potassium chlorides (KCl) were deposited on the substrate but not on the film after peeling (Supplementary Fig. 9). The deposited potassium chlorides (KCl) were easily cleaned off by water scrubbing.”.
3. Supplementary Fig. 9 in Supplementary file:

Supplementary Fig. 9. Photomicrograph and EDS characterization of the substrate/film after thin film detachment ($U = 4.5$ V). a Photomicrograph of the substrate after thin film detachment. The dendrite line represents the potassium

chloride (KCl) precipitation on the substrate. **b** Optical image of the detached film on the substrate after water scrubbing. **c-d** EDS characterization of the substrate and the PDMS film, respectively. Some potassium chlorides (KCl) are deposited on the substrate (about 1.2 wt%) but not on the films. The EDS results show no discernible differences between the substrate/film before peeling and after the water scrubbing.

4) The authors should explain why Figure 2e stops at 80s.

Response: We greatly appreciate the referee's constructive comments. We used a single liquid droplet of 20 μL in the experiments to evaluate the electro-capillary peeling behavior. Liquid droplets of such volume would completely spread in some film bonding layers at about 80 s. Therefore, the peeling data were collected within 80 s to ensure the concordance of the experiment. In the revised manuscript, we added a detailed description to explain why the peeling length data were collected within 80 s.

Additionally, we proposed a long-term detachment strategy by replenishing liquid during the electro-capillary peeling (as discussed in the response in comments 2 and 3). Exploiting this strategy, we conducted ten-minute electro-capillary peeling experiments with thin films of 1.0, 1.6, and 2.4 MPa to describe the long-term peeling behavior. The revision in the revised manuscript is as follows.

1. In the legend of Figure 2 (Lines 125-126): "The peeling length data are collected within 80 s because a liquid droplet of 20 μL would completely spread in some bonding layers at this time."
2. In the Discussion section (Lines 275-280): "To evaluate the electro-capillary peeling method in long-term detaching, a strategy of replenishing liquid during the electro-capillary peeling was proposed for large-area film detachment (Supplementary Fig. 7). Exploiting this strategy, a thin film with a diameter of 5 cm was entirely peeled off at about 8.3 min (Supplementary Fig. 8). The detached thin film could be picked up and transferred to another substrate (Supplementary Fig. 8). Statistical results of peeling length revealed that the electro-capillary peeling method had a stable detaching rate in the long-term detachment when the electrolyte solution was adequate."

5) Despite the experimental data correspond pretty well with the scaling law, I still encourage the authors to use the experimental data to determine the term “ f_{adhesion} ” and discuss whether it is a consistent value over all the samples being used. It is supposed to have a constant value for a specific interface regardless of the coating thickness and coating elastic modulus. If not, the authors should discuss the discrepancy.

Response: We thank the referee for the constructive comments. As discussed in our response to comment 1, we used the water blister method to measure the PDMS films’ wet adhesion with various thicknesses (from 25 to 300 μm) and elastic moduli (from 1.0 to 2.4 MPa) in the revised manuscript. The measurement results presented that the wet adhesion of PDMS film on the ITO glass was almost the same (about 117 mJ m^{-2}) regardless of the thin film thickness and elastic modulus (Supplementary Tables 1 and 2). After disregarding experimental errors, the wet adhesion of the bonding layer approximately possessed a constant value for the PDMS film on the ITO glass, which was independent of the thin film thickness and elastic modulus. The revision in the revised manuscript is as follows.

1. In the Results section (Lines 136 and 138): “The water blister test shown in Supplementary Fig. 2 and Supplementary Tables 1-2 showed that the wet adhesion of these films on the ITO glass was almost the same regardless of the thin film thickness and elastic modulus.”.

2. Supplementary Tables 1 and 2 in Supplementary Information:

Supplementary Table 1. The wet adhesion of PDMS films (1.6 MPa) with different thicknesses.

Thickness (μm)	Wet adhesion (mJ m^{-2})
25	115 ± 21
50	108 ± 33
100	122 ± 38
200	115 ± 59
300	125 ± 26

Supplementary Table 2. The wet adhesion of PDMS films (100 μm) with various elastic moduli.

Elastic modulus (MPa)	Wet adhesion (mJ m^{-2})
1.0	135 ± 36
1.6	122 ± 38
2.4	112 ± 16

We thank the referee again for the kind help, constructive comments, and insightful suggestions. We are lucky to have the referee who provided very insightful and constructive comments and suggestions that fundamentally improved the scientific depth of our work. We believe that with the referee's help, the revised manuscript is suitable for publication in *Nature Communications* now.

Authors' Responses to Reviewer #4

The manuscript introduces a novel method for the detachment of thin films from substrates called the electro-capillary peeling method. Unlike traditional techniques, this method uses an electric field to drive a liquid into the bonding layer, allowing the film to detach. This is the first example of using an electric field for this purpose. This method is effective across a wide variety of films and voltages, showing that it is an active control technique. The detachment process is driven by Maxwell stress in the electric field, which competes with the film's adhesive and tension stresses. The authors also provide mathematical relationships for peeling length with respect to time, applied voltage, film thickness, and elastic modulus. The film's deformation under this method showed a significantly lower strain, making it suitable for protecting functional devices. The study further reveals that a relatively low critical peeling voltage can be used to detach films across various solutions of different concentrations. Overall, this paper is of high quality, and I recommend the publication of this paper after addressing the following points.

Response: We thank the referee for the time spent on our paper and the constructive comments. We are pleased that the referee considers our work to be “high quality”. Empowered by the referee’s concern about the organic solvents’ applications in the electro-capillary peeling method and the deformation of the detached films, we conducted additional experiments and provided convincing experimental details in the revised manuscript to address these queries. We believe that our paper’s scientific merit is much improved after revising following the referee’s comments. The details of the changes that we made are shown in the following response, and revisions of the text are highlighted in the text body.

- Some materials are vulnerable to water, which can induce the oxidation or dissolution of some films. I am curious if other organic solvents can also be used.

Response: We thank the referee for this constructive comment. In the revised manuscript, we performed additional experiments to evaluate the organic solvents’

applications in the electro-capillary peeling method. The observation showed that the electro-capillary peeling method could be applied in the organic solvents when they dissolved some acids or salts, including ethanol, acetone, and glycerol solution (Supplementary Fig. 13). Thus, organic solvents could be applied in the electro-capillary peeling method to detach a thin film vulnerable to water. The revision in the revised manuscript is as follows.

3. In the Discussion section (Lines 301-303): “For some thin films vulnerable to water, the electro-capillary peeling method could be applied to detach them using organic solvents, such as alcohol, acetone, and glycerol solutions (Supplementary Fig. 13).”.

4. Supplementary Fig. 13 in Supplementary file:

Supplementary Fig. 13. Electro-capillary peeling method using different types of organic solvents. **a** Schematic diagram of electro-capillary peeling applied with organic solvents. **b** The detaching process of the electro-capillary peeling method applied with organic solvents. The tested organic solvents are (±)-Camphor-10-sulfonic acid/alcohol, (±)-Camphor-10-sulfonic acid/acetone, and sodium hydroxide/glycerol solutions. The concentrations of solutions are 0.5 mol L^{-1} , and the scale bars are 2 mm.

- The application of voltage on the water droplet can facilitate the electrochemical corrosion of the film. A high voltage would be needed for tightly bonded films to initiate detachment. Also, some films will be dissolved into water, and then the dissolved ions can be deposited somewhere on the substrate. I wonder if there are any additional test results regarding this point.

Response: We thank the referee for the constructive comment. In the revised manuscript, we conducted the photomicrograph and EDS characterization to investigate this issue. The observation showed that there were only some potassium chlorides (KCl) deposited on the substrate after electro-capillary peeling (Supplementary Fig. 9). In the experiment, it was not found the film dissolved into the water. We considered this finding related to our experimental samples being PDMS films, which always had stable properties. For some water-soluble films, there may be a finding of the film dissolving in water. In these cases, organic solvents could be applied in the electro-capillary peeling to detach these films according to the referee's suggestions (as discussed in our response to comment 1). The revision in the revised manuscript is as follows.

1. Supplementary Fig. 9 in the Supplementary file:

Supplementary Fig. 9. Photomicrograph and EDS characterization of the

substrate/film after thin film detachment ($U = 4.5$ V). **a** Photomicrograph of the substrate after thin film detachment. The dendrite line represents the potassium chloride (KCl) precipitation on the substrate. **b** Optical image of the detached film on the substrate after water scrubbing. **c-d** EDS characterization of the substrate and the PDMS film, respectively. Some potassium chlorides (KCl) are deposited on the substrate (about 1.2 wt%) but not on the films. The EDS results show no discernible differences between the substrate/film before peeling and after the water scrubbing.

2. In the Discussion section (Lines 280-284): “In addition, we investigated the characterization of the substrate and thin film after electro-capillary peeling (Supplementary Figs. 9 and 10). The photomicrograph and energy dispersive spectroscopy (EDS) results showed that some potassium chlorides (KCl) were deposited on the substrate but not on the film after peeling (Supplementary Fig. 9). The deposited potassium chlorides (KCl) could be cleaned off by water scrubbing.”.

- As the detachment proceeds, the distance between the Pt wire and the very interface where the detachment occurs will change. So, I believe a more dynamic system should be developed to maintain a constant detachment speed.

Response: We are grateful for the referee’s constructive comment. In the revised manuscript, we conducted ten-minute electro-capillary peeling experiments with thin films of 1.0, 1.6, and 2.4 MPa to describe the long-term detachment behavior. The acquired experimental result showed that, although the peeling distance increased, the detaching speed of the electro-capillary peeling was almost a constant value (Supplementary Fig. 7). The working mechanism demonstrated that the detachment speed of the electro-capillary peeling method was $v \sim \varepsilon_f^3 \rho_f^3 \zeta^2 U / (\mu E^2 d_0^2)$, which was independent of the peeling distance in a solution with uniform electrical properties. Due to the position-dependent electrical properties, the detachment speed would be influenced by the peeling distance in a solution with uneven charge density. In addition, it can be inferred from the working mechanism that the electro-capillary peeling detachment speed can be actively controlled by the applied voltage. Therefore,

for a solution with uneven charge density, a dynamic system could be developed by altering the applied voltage in response to the charge density change to maintain a constant detachment speed. The revision in the revised manuscript is as follows.

1. Supplementary Fig. 7 in Supplementary file:

Supplementary Fig. 7. A long-term detachment strategy of the electro-capillary peeling method. **a** Schematic illustration of the long-term detachment strategy. A micropump is used to replenish fluid during the electro-capillary peeling. **b** Side and top views of the long-term peeling process. Black, red, and blue boxes are used to mark the thin film detachment with elastic moduli of 1.0, 1.6, and 2.4 MPa, respectively. The tested time is 660 s, and the scale bars in the side and top views are 1 cm. **c** Statistical results of peeling length in the long-term peeling. The thickness of tested films in **b** and **c** are 100 μm , and the error bars in **c** are the standard deviation of the raw data.

2. In the Discussion section (Lines 279-280): “Statistical results of peeling length revealed that the electro-capillary peeling method had a stable detaching rate in

the long-term detachment when the electrolyte solution was adequate.”.

- More demonstration of detached films is needed. I expect there will be deformation of detached films after transferring them onto other substrates.

Response: We thank the referee for the insightful and constructive comment. In the revised manuscript, we explored the deformation of detached films in both the horizontal (xoy plane) and vertical (z -direction) directions (Supplementary Fig. 10). The observation in the horizontal direction showed that the positions of the PDMS film’s inner and outer boundaries were not out of shape after transferring to another substrate. The scanning results in the z -direction presented that the PDMS film thickness was nearly identical before and after peeling. Thus, the thin film had no residual deformation after peeling and transferring. The revision in the revised manuscript is as follows.

1. Supplementary Fig. 10 in the Supplementary file:

Supplementary Fig. 10. Deformation of detached films after transferring. a Side

and top views of PDMS films' three-dimensional topography. r_1 and r_2 are the inner and outer boundaries' radius of the thin film, respectively, and routes 1, 2, 3, and 4 are the surface profiler scan paths. **b** The position of the PDMS film's inner and outer boundaries before and after transferring. Black square dots and the orange solid line represent the position of the boundaries after and before transferring, respectively. **c** The z -coordinate of PDMS films' thickness before and after transferring. z and l represent the z -coordinate and the scan length.

2. In the Discussion section (Lines 284-286): “After transferring detached films onto other substrates, the thin film profile was consistent with the initial state and had no residual deformation (Supplementary Fig. 10).”.

3. In the Methods section (Lines 375-380):

“Characterization of the thin film profile. The thin film profile before and after peeling was collected using a probe step profiler (Dektak XT, Bruker, Germany) with a 10 μN force on the microneedle (12.5 μm in diameter). The tested thin films had a diameter of 30 mm and a thickness from 25 μm to 300 μm . As shown in Supplementary Fig. 10, two profile parameters were tested by the probe step profiler, including the contour lines of the inner and outer circle boundary in the xoy plane and the z -coordinate of the top surface. The z -coordinate of the top surface was collected from 4 scan paths.”.

- In order to verify the integrity of the detached thin films, additional device test results would be needed. (Data in Figure 5d are not sufficient.)

Response: We thank the referee for the constructive comments. In the revised manuscript, we used SEM, AFM, and probe step profiler to verify the integrity of the detached thin films. For the thin film without micro/nanostructure, a probe step profiler was used to explore the deformation of the detached films in both the horizontal (xoy plane) and vertical (z -direction) directions (as discussed in our response to comment 5). The observation demonstrated that the thin film had no residual deformation after transferring (Supplementary Fig. 10). For the thin film implanted with ZnO nanorods, SEM and AFM were utilized to investigate the

integrity of nanolayers and the size of the cracks. The observation showed that the thin film's ZnO nanolayer occurred many cracks (about 2 μm in width and 10 μm in height) after being detached by the debonded strip approach but maintained intact after being peeled off by the electro-capillary peeling method (Supplementary Fig. 6). These tested results enrich the data shown in Fig. 5d, displaying the change of ZnO nanolayer on the film after detaching by different approaches. The revision in the revised manuscript is as follows.

1. Figure 5d in the Results section:

Fig. 5| Deformation of the film during the electro-capillary peeling process. a-b Displacement and strain fields of the film detached using the electro-capillary peeling method. The displacement and strain fields exhibit deformation in the z-direction. Colour bars represent the displacement and strain values. c Strain of film versus peeling rate in the various methods. The debonded strip method peels the film at an angle of 90°. The thickness of the tested films is 100 μm . d **Characterization of the ZnO nanorods on the film after peeling ten times.** The ZnO nanolayer occurs many cracks when a thin film is detached by the debonded strip but maintains intact when peeled off by the electro-capillary peeling method. Scale bars are 10 μm .

2. Supplementary Fig. 6 in the Supplementary file:

Supplementary Fig. 6. SEM and AFM tests of ZnO nanolayer on the film. Black, red, and blue boxes mark three groups of ZnO nanorods characterization. The control group is the ZnO nanolayer on the film without peeling. The insert photo is the AFM result of the crack on the ZnO nanolayer.

We thank the referee again for the kind help and constructive comments. We are lucky to have the referee who provided very insightful and constructive comments and suggestions that fundamentally improved the scientific depth of our work. We believe that with the help of the referee, the revised manuscript is suitable to be published in *Nature Communications* now.

REVIEWERS' COMMENTS

Reviewer #3 (Remarks to the Author):

The authors have addressed all my comments with well-designed experiments and thorough discussions. I do not have further comments and I think the work should be published as it is.

A minor comment: The equation (10) cited by the authors has a typo. In the original paper by Sofla et al., $n(v)=0.809 - 1.073v - 0.816v^2$. The correct expressions should be $n(v)=0.809 + 1.073v - 0.816v^2$. I discussed this issue with Dr. Matthew R. Begley (the corresponding author of this paper) back in 2018. The mistake will cause deviations in blister test when film thickness is comparable to blister height.

As thin film adhesion is a key concept in the paper, I encourage the authors to re-calculate the adhesion based on the corrected equations. But I do not need to go through the revision again. Overall, this is a very nice work and I want to congratulate the authors for their accomplishments.

Reviewer #4 (Remarks to the Author):

I carefully checked the revised manuscript and confirmed that the authors successfully addressed my previous concerns. I now agree with the publication of the paper.

中国科学院力学研究所非线性力学国家重点实验室

State Key Laboratory of Nonlinear Mechanics (LNLM)

Institute of Mechanics, Chinese Academy of Sciences (CAS)

September 10, 2023

General Description for Reviewers

I, along with my coauthors, would like to submit the revised manuscript entitled “**Electro-capillary peeling of thin films**” for publication in *Nature Communications*.

Manuscript ID: NCOMMS-23-07640A

Title: Electro-capillary peeling of thin films

We thank the reviewers again for the time spent reading our manuscript, the recognition of our work, and the constructive comments and suggestions, which are very helpful in improving our paper. After carefully reading the reviewers’ comments, we diligently made efforts to address these queries in the revised manuscript. These revisions made great improvements to our manuscript that will be suitable for publishing in *Nature Communications*.

With my best regards.

Sincerely yours,

Professor Ya-Pu Zhao (corresponding author)

State Key Laboratory of Nonlinear Mechanics

Institute of Mechanics, Chinese Academy of Sciences

Beijing 100190, CHINA

Phone: 0086-10-82543932

E-mail: yzhao@imech.ac.cn

Authors' Responses to Reviewer #3

The authors have addressed all my comments with well-designed experiments and thorough discussions. I do not have further comments and I think the work should be published as it is.

Response: We thank the referee for the thorough review of our revised manuscript and are grateful for the positive comment. We are pleased to hear that we have successfully addressed all of the previous concerns and the paper should be published in Nature Communications as it is. The typo in equation (10) was modified in the revised manuscript, and the details of the changes we made are shown in the following response and highlighted in the text body.

A minor comment: The equation (10) cited by the authors has a typo. In the original paper by Sofla et al., $n(v)=0.809 - 1.073v - 0.816v^2$. The correct expressions should be $n(v)=0.809 + 1.073v - 0.816v^2$. I discussed this issue with Dr. Matthew R. Begley (the corresponding author of this paper) back in 2018. The mistake will cause deviations in blister test when film thickness is comparable to blister height.

As thin film adhesion is a key concept in the paper, I encourage the authors to re-calculate the adhesion based on the corrected equations. But I do not need to go through the revision again. Overall, this is a very nice work and I want to congratulate the authors for their accomplishments.

Response: We thank the referee for the helpful comments. In the revised manuscript, we modified equation (10) following the referee's advice and re-calculated the adhesion based on the corrected equations. The measured wet adhesion was updated in the revised manuscript. The revision in the revised manuscript is as follows.

1. In the Methods section:

$$\begin{cases} m(v) = 0.509 + 0.221v - 0.263v^2, \\ n(v) = 0.809 + 1.073v - 0.816v^2. \end{cases} \quad (10)$$

2. Supplementary Tables 1 and 2 in Supplementary Information:

Supplementary Table 1. The wet adhesion of PDMS films (1.6 MPa) with different

thicknesses.

Thickness (μm)	Wet adhesion (mJ m^{-2})
25	130 ± 25
50	122 ± 38
100	142 ± 43
200	129 ± 66
300	146 ± 28

Supplementary Table 2. The wet adhesion of PDMS films (100 μm) with various elastic moduli.

Elastic modulus (MPa)	Wet adhesion (mJ m^{-2})
1.0	153 ± 40
1.6	142 ± 43
2.4	126 ± 23

Again, we thank the referee for the time, effort, and expertise in reviewing our manuscript and are grateful for your support throughout this process.

Authors' Responses to Reviewer #4

I carefully checked the revised manuscript and confirmed that the authors successfully addressed my previous concerns. I now agree with the publication of the paper.

Response: We thank the referee for the thorough review of our revised manuscript and are grateful for the positive comment. We are pleased to hear that we have successfully addressed all of the previous concerns, and the paper is agreed to be published.

Again, we thank the referee for the time, effort, and expertise in reviewing our manuscript and are grateful for your support throughout this process.